

# Challenges and opportunities for Arabic question-answering systems: current techniques and future directions

Asmaa Alrayzah[1,2], Fawaz Alsolami[1] and Mostafa Saleh[1]

[1] Faculty of Computing and Information Technology, King Abdulaziz University, Makkah, Jeddah, Saudi Arabia
[2] College of Computer Science and Information Systems, Najran University, Najran, Najran, Saudi Arabia

## ABSTRACT

Artificial intelligence-based question-answering (QA) systems can expedite the performance of various tasks. These systems either read passages and answer questions given in natural languages or if a question is given, they extract the most accurate answer from documents retrieved from the internet. Arabic is spoken by Arabs and Muslims and is located in the middle of the Arab world, which encompasses the Middle East and North Africa. It is difficult to use natural language processing techniques to process modern Arabic owing to the language's complex morphology, orthographic ambiguity, regional variations in spoken Arabic, and limited linguistic and technological resources. Only a few Arabic QA experiments and systems have been designed on small datasets, some of which are yet to be made available. Although several reviews of Arabic QA studies have been conducted, the number of studies covered has been limited and recent trends have not been included. To the best of our knowledge, only two systematic reviews focused on Arabic QA have been published to date. One covered only 26 primary studies without considering recent techniques, while the other covered only nine studies conducted for Holy Qur'an QA systems. Here, the included studies were analyzed in terms of the datasets used, domains covered, types of Arabic questions asked, information retrieved, the mechanism used to extract answers, and the techniques used. Based on the results of the analysis, several limitations, concerns, and recommendations for future research were identified. Additionally, a novel taxonomy was developed to categorize the techniques used based on the domains and approaches of the QA system.

## INTRODUCTION

In recent years, the capacity of computing systems to comprehend natural languages has increased and several computer-based programs, such as question-answering (QA) systems, allow individuals to expedite their work. A QA system is a subfield of information retrieval (IR) and is considered as an alternative to search engines. The objective of a QA system is to provide correct answers to inquiries submitted by users using their natural

Corresponding author
Asmaa Alrayzah,
asalrayzah@nu.edu.sa

language. Conversely, search engines return related documents to extract relevant answers (*Shaheen & Ezzeldin, 2014*).

QA systems read passages and answer questions posed by users in their natural language. Given a question, they extract and rank possible answers from the most relevant documents retrieved from the internet to determine the most accurate answer. However, training these systems to understand Arabic text and answer questions posed in Arabic is a challenge. Arabic words often exhibit polysemy, meaning they can have multiple meanings depending on the context; for instance, consider the Arabic word "ذهب", which can refer to "gold", but can also mean "he went", adding ambiguity to its usage in various contexts. This semantic richness poses difficulties for NLP models, necessitating advanced techniques to disambiguate and achieve accurate understanding. Over the years, numerous QA systems in various languages have been designed for usage. However, the development of Arabic QA systems has been impeded by the scarcity of research resources, tools, and linguistic challenges in Arabic (*Darwish et al., 2021*). Given these challenges, building QA systems that can understand and correctly respond to Arabic questions is highly difficult. As a result, very few studies on Arabic natural language processing (NLP) are available compared to those on English NLP. Therefore, computer-based analysis and comprehension of Arabic text have recently emerged as a burgeoning NLP research area.

Previously, a few Arabic QA experiments and systems were designed using datasets that were small or some inaccessible. Despite several reviews being conducted on Arabic QA studies, either the number of studies covered was limited or they failed to encompass the latest trends in research (*Alwaneen et al., 2021*; *Bakari, Bellot & Neji, 2016c*; *Bakari, Bellot & Neji, 2016b*; *Biltawi, Tedmori & Awajan, 2021*; *Ezzeldin & Shaheen, 2012*; *Mahdi, 2021*; *Ray & Shaalan, 2016*; *Utomo, Suryana & Azmi, 2020*). To our knowledge, only two published systematic reviews were focused on Arabic QA systems. The first review covered only 26 primary studies without including recent techniques (*Biltawi, Tedmori & Awajan, 2021*), while the other covered only nine studies conducted for the Holy Qur'an QA systems (*Utomo, Suryana & Azmi, 2020*). The studies that were incorporated were examined based on the dataset employed, domain addressed, type of Arabic questions, IR and answer extraction mechanisms, and techniques utilized. Additionally, a new taxonomy was developed for the approaches used, which were categorized according to both the QA system domains and methods.

This systematic review is structured as follows. 'Background' provides a background on the challenges of the Arabic language, history of NLP, and approaches to Arabic QA tasks. 'Related work' presents an overview of previous research in this area. 'Survey methodology' provides a detailed description of the methods used to conduct this review. The findings are presented in 'Results'. The limitations of this review are discussed in 'Limitation'. Lastly, 'Conclusions' draws the conclusions of this study.

## BACKGROUND

### History and approaches of QA systems in the Arabic world

Currently, almost everything is just a click away and progress is rapid in this world. A fundamental Google search engine command includes "what will today's weather be like? You may need an umbrella; the temperature will be 8 °C (rain)." Artificial intelligence (AI) has taken great strides in recent years and is now an integral part of numerous aspects of our lives. Researchers on AI have improved research on NLP through their ongoing efforts. NLP was initially proposed to help computers better understand human languages. Some of the earliest applications of AI were in the field of NLP, such as machine translation (*Khurana et al., 2022*). An exponential improvement in the quality of AI technology, accompanied by the growing public knowledge and expectations of what AI can accomplish were observed in recent years (2010–2020).

According to *Darwish et al. (2021)*, Arabic NLP and QA have especially undergone three distinct waves of development over their history. The fourth wave of development can be accelerated based on the approaches of NLP. Four of the approaches include rule-based (RB), machine learning (ML), deep learning (DL), and pre-trained modelling languages (PMLs), which are also known as transformer models based on DL.

In the 1980s, the world witnessed a significant breakthrough when Microsoft released MS-DOS 3.3 with Arabic language support. Additionally, Sakhr (*Darwish et al., 2021*) developed the first Arabic morphological analyzer in 1985, where examining the structure of Arabic text through morphological analysis was emphasized, and a majority of the research utilized RB methods. Moreover, Sakhr (*Darwish et al., 2021*) developed the first syntactic and semantic analyzer in 1992, followed by Arabic optical character recognition in 1995. Numerous commercial Arabic-to-English machine translation products and solutions were developed during that era (*Khurana et al., 2022*). The wave-processing Arabic language was based on the RB approach, where the handwritten rules were utilized to assist machines in comprehending sentences by structuring phrases. However, the technique did not enable a machine to grasp the meaning of a sentence, but it could recognize particular words or word combinations in specific patterns.

During 2000–2010, the second wave of Arabic NLP development occurred. Large-scale initiatives, funded by the United States government, were conducted to develop Arabic NLP tools for their dialects. These initiatives included the creation of machine translation technology, IR systems, and QA tasks (*Darwish et al., 2021*). A majority of the systems then developed employed ML, which was gaining immense popularity in the field of NLP. ML-based algorithms require significantly less linguistic information than RB-based systems, and are hence more efficient and accurate. However, acquiring requisite data requires some effort, particularly for Arabic datasets. During this period, numerous hybrid systems that successfully merged RB morphological analyzers and ML disambiguation were developed (*Darwish et al., 2021*). ML-based algorithms offered a more sophisticated method of interpreting ambiguity. Decision trees as examples of algorithms employed if-then regulations to derive the most accurate outcome, while probabilistic algorithms reinforced the decision of machines by indicating a degree of certainty (*Johri et al., 2021*).

In 2010, the third wave of development based on the DL approach was initiated (*Alyafeai, AlShaibani & Ahmad, 2020*). During this period, a noteworthy surge in the quantity of Arab researchers and postgraduate students interested in Arabic NLP was observed, which resulted in a corresponding rise in publications from the Arab world presented at top conferences. Additionally, two major independent advancements emerged during this period: DL and neural models and social media. NLP involved inherent ambiguity that could not be resolved entirely. The meaning of a word could depend on the context in which the word was used, making it challenging to create a definitive rule or decision tree that covered all possible meanings. Since DL did not necessitate the programmer to provide decision-making rules, but rather had an algorithm deduce the process of mapping an input to an output, it was an effective solution to the existing issue (*Johri et al., 2021*). A majority of the artificial neural networks (ANN), including recurrent neural networks (RNNs) and conventional neural networks (CNNs), were introduced for NLP. However, to train and test DL models, high-power hardware with vast processing speed was necessary. Further details are discussed in the upcoming sections.

NLP requires a large amount of labeled data for training models. One of the main challenges in NLP is obtaining sufficient labeled data, which are not always readily available, making it difficult to train transformer models (*Johri et al., 2021*). A method to overcome this hurdle is to label data explicitly, although the process may be time-consuming and expensive.

In 2018, the fourth wave of development started by the implementation of mBERT (*Pires, Schlinger & Garrette, 2019b*). A majority of the NLP tasks in this period was solved by pre-trained language models (PLMs), also known as transformer models. Transformer models are DL models employing an attention mechanism. A self-attention mechanism does not require recurrent architecture and can perform parallel processing (*Zong, Xia & Zhang, 2021*). It prevents the loss of relevant information from the extensive volume of texts processed by ANN models. Thus, the mechanism is an integral part of models that perform several NLP tasks including translation, QA, and sentiment analysis (*Saidi, Jarray & Mansour, 2021*). It allows dependencies to be processed regardless of their position in the input or output sequences, which is necessary for NLP because a transfer model searches an encoder for positions containing the most relevant information to generate a sentence. Thus, a transfer model is capable of "attending to" specific words when encoding an output because of the self-attention mechanism, which allows it to remember all the tokens in the input sequence (*Vaswani et al., 2017*). Table 1 provides a summary of approaches in NLP.

## Challenges in Arabic NLP

Arabic is spoken by the Arabs and Muslims and is located in the middle of the Arab world, which encompasses the Middle East and North Africa. The Arabic language is one of the oldest and most widely spoken languages in the world, with over 420 million speakers across the globe (*Darwish et al., 2021*). The use of NLP techniques to process modern Arabic is difficult due to the complex morphology, orthographic ambiguity, regional variations

Alrayzah et al. (2023), *PeerJ Comput. Sci.*, DOI 10.7717/peerj-cs.1633

**Table 1  NLP approaches.**

|  | RB | ML | DL (CNN/RNN) | DL (PLMs/Transformer) |
|---|---|---|---|---|
| Year | (1985–2000) | (2000–2010) | (2010–2020) | (2018–Present) |
| Concept | Linguists write explicit rules such as the if-then rule. These human-made rules are followed and applied to store, sort, and manipulate data. | Humans extract features. Thereafter, machines learn the rules based on extracted features, such as decision-tree algorithms. | Machines learn the rules and features without considering contexts, such as ANN and CNN models. | Machines learn rules and features with self-attention mechanism to consider contexts, such as transformer models. |
| Pros | Rules are easy to understand. Requires limited data owing to limited applicable domain. | Use of fewer computational resources and less data. | Automatic representation of learning and permission to capture semantic. | Existence of self-attention mechanism and permission to consider context. Texts are processed in parallel. |
| Cons | Manual performance of task is time consuming and difficult to scale. When rules are not strictly followed, performance can suffer. | No semantic capturing, feature extraction is expensive, and no proper generalization for other tasks, such as text generation. | Sequential processing of texts. Requires substantial quantity of data and expensive computing power. | Requires substantial quantity of data and expensive computational resources. |

while speaking, and limited linguistic and technological resources of the language (*Alyafeai, AlShaibani & Ahmad, 2020*; *Darwish et al., 2021*; *Guellil et al., 2021*).

### Morphological richness

In Arabic, a single word can have multiple meanings; for instance, "سائل" can mean either "liquid" or "beggar." Further, a single word may be equivalent to an entire English phrase; for example, "فسينفقونها" means "they will spend it." Emmett Knowlton (*Atef et al., 2020*) has stated that Arabic is regarded as one of the most difficult languages to master, with an average of 88 weeks or 2,200 h of instruction needed to attain proficiency in both spoken and written Arabic.

### Orthographic ambiguity

Written Arabic texts use optional diacritical marks to indicate details of phonology and vowels that are essential for distinguishing one word from another, resulting in orthographic ambiguity (*Darwish et al., 2021*). For example, the word "كتبت" is written without diacritics; however, it can be written as "كَتَبْتُ or I wrote," "كَتَبَتْ or she wrote," and "كُتِبَتْ or it is written by" with diacritics.

### Dialectal variations

Arabic is not a singular language, but a group of linguistically related varieties, including modern standard Arabic (MSA), classical Arabic (CA), and Arabic dialects (AD) (*Guellil et al., 2021*; *Malhas, 2023*). MSA is preferred for official purposes and education, while other forms including AD are commonly spoken and are recently being used for written communication. AD encompasses various dialects, such as Egyptian, Gulf Arabic, or Moroccan Arabic, each with their own unique grammar and vocabulary that distinguishes them from MSA. For example, the word "car" in MSA, Egyptian, Gulf Arabic, and Moroccan Arabic is written as "سيارة" "عربية", "موتر", and "كرهبة," respectively (*Darwish et al., 2021*).

### Resource poverty

Arabic has one of the lowest resources of data. Substantial quantity of Arabic data has to be collected and processed through several pre-processing steps before an input can be given to a machine for learning. The data utilized in RB methods necessitate lexicons and meticulously written rules, while those in ML and DL approaches require large and annotated corpora.

Unlike other languages, Arabic is not usually supported by various platforms. *Miller (1995)* created an English lexical database of words known as WordNet to store semantic relations between words. Thereafter, several other languages were incorporated except Arabic. WordNet research was supported by Princeton University to enhance English words and their relations. In 2006, the development of Arabic WordNet was initiated, and was improved and processed until 2016 by *Elkateb et al. (2006)*.

In 2002, Sheffield University created a general architecture for text engineering, an NLP tool for several tasks such as information extraction (*Cunningham, 2002*). Initially, it did not support Arabic, but since 2010, *Zaidi, Laskri & Abdelali (2010)* endorsed it. Up to

this date, Arabic is yet to be supported by models including those researched by Fan and Gardent (*Fan et al., 2020*), who built a model aimed at generating texts for 21 languages. Conversely, Arabic does not support multilingual abstract meaning representation-to-text generation models.

## Arabic QA systems

NLP is a linguistic branch of AI concerning the ability of computing systems to comprehend natural languages and perform specific tasks (*Prasad et al., 2019*). Prominent research areas in NLP include QA, sentiment analysis, translation, and computer-based text generation. Computers employ various approaches, such as lexicon RB and ML approaches, to process, analyze, and comprehend natural languages. Transformer models form the basis of current NLP techniques that yield cutting-edge outcomes (*Otter, Medina & Kalita, 2021*; *Al-Ayyoub et al., 2018*).

With the advancement of technology and availability of ample quantity of online data, the ability to request information is increasingly important. The rapid expansion of online information is a crucial draw for several users who depend on search engines and other IR tools to find answers to their questions. When a user enters a request into a search engine, the engine scours the internet for relevant pages and returns a list of those with brief descriptions of the request (*Calijorne Soares & Parreiras, 2020*). Consequently, QA systems are crucial to the fields of IR and text processing because they facilitate the extraction of important information from a given text (*Almotairi & Fkih, 2021*). A QA system is one of the NLP tasks belonging to AI tasks. The field of a QA system may be regarded as a subset of natural language understanding (NLU), which is a subset of NLP. Automatic QA systems are created to respond to questions presented by individuals using natural languages. The users input questions in their own languages, extract crucial details from the provided data, and receive responses in their natural language format (*Nassiri & Akhloufi, 2022*).

The purpose of a QA system is to analyze questions or queries posed in natural languages by users and provide the most relevant answers to them (*Nguyen & Tran, 2022*). Hence, QA systems are considered as intelligent systems owing to their goal of providing precise answers to user questions posed in natural languages. Moreover, QA systems are vital for improving particular field environments such as education, health care, and research engines. They are more efficient in promptly providing correct answers to user questions (*Almotairi & Fkih, 2021*).

QA systems find applications in numerous domains, such as education, health care, research engines, and personal assistance, for various reasons (*Calijorne Soares & Parreiras, 2020*). Due to their versatility in various applications, QA systems are commonly utilized in the field of NLP (*Almotairi & Fkih, 2021*). They involve QA using natural languages. Several examples of such applications are as follows.

First, the main application of a QA system is in IR or web search (*Calijorne Soares & Parreiras, 2020*). QA systems automatically answer user questions in their natural languages and do not merely provide documents relevant to the questions; they also extract all relevant information from those documents and present a thorough response, similar to that a human would.

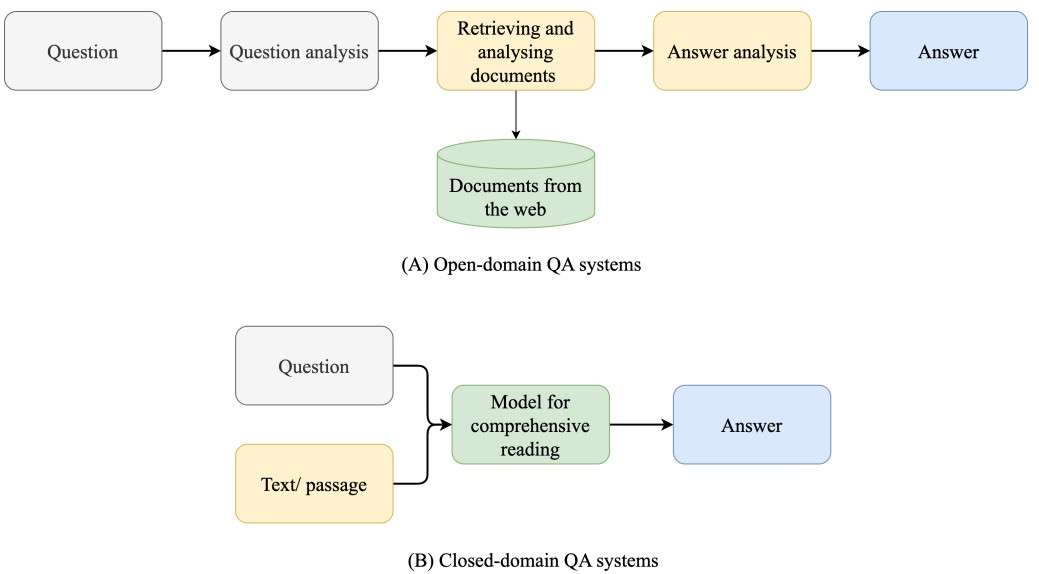

(A) Open-domain QA systems

(B) Closed-domain QA systems

**Figure 1  Open and closed-domain QA systems.**

Second, QA systems are used in community QA (CQA) services (*Adlouni et al., 2019*), where users converse and share information through QA, such as Yahoo! answers, stack overflow, or Cheeg, aided by online knowledge exchange. CQA services are used in the field of education to help learners interact with experts.

Third, QA systems are used commercially as customer support to reduce the workload on customer service teams and allow them to concentrate on important issues. An example of a QA system applied to customer support is the chatbot used by Amazon (*Jiang, 2019*).

Fourth, QA systems are used by search engines such as MSN, Google, Yahoo!, and Bing Search (*Al-Shenak, Nahar & Halawani, 2019*). The number of displayed questions grows when either of the questions is clicked, and the revised list more closely resembles the clicked question.

Lastly, a QA system is used for real time QA (*Elfadil, Jarajreh & Algarni, 2021*). Even the most complicated questions have to be answered within a few seconds, as users would face an unpleasant experience of spending hours in front of a computer awaiting answers. Therefore, the development of QA systems providing end products in real time is highly necessary. They are required in every aspect involving assistance from computers.

QA systems are categorized into two: reading comprehension (RC) and IR systems. RC systems are capable of reading and understanding text passages and answering related questions (*Zhu et al., 2021*). RC is used as a metric to evaluate the ability of a computing system to understand human language, and has several practical applications. RC systems are given specific passages as inputs, while IR systems must extract answers from an extensive collection of web documents without prior knowledge of their location (*Zhu et al., 2021*). Thus, QA systems are of two main types: open-domain and close-domain, as shown in Fig. 1. Open-domain QA systems access several documents on the internet as input and retrieve answers from them (*Alsubhi, Jamal & Alhothali, 2021*). In contrast,

close-domain QA systems retrieve answers from specific passages by taking both the question and passage as input.

## RELATED WORK

Thus far, several review papers have discussed Arabic QA systems, resources, tools, and techniques, while focusing on the RB approach. A summary of these review papers is presented in Table 2, with data categorization based on the review type, publication year, years covered by the studies, number of both Arabic QA studies and datasets included, and whether Arabic QA taxonomy, DL models, and evaluation methods are present. Additionally, limitations of those studies are included. Moreover, a comparison between our research and existing publications is also included in Table 2.

Data presented in Table 2 suggest that only a few, approximately nine (*Utomo, Suryana & Azmi, 2020*) to 26 (*Biltawi, Tedmori & Awajan, 2021*) Arabic QA studies were conducted by researchers because a majority of these surveys and review papers focused on Arabic QA systems developed using the RB approach, particularly for open-domain purposes, which was most prevalent in general domain for Arabic QA frameworks. *Biltawi, Tedmori & Awajan (2021)* conducted a systematic literature review (SLR), which encompassed the most extensive collection of Arabic QA studies, spanning several recent years. However, *Biltawi, Tedmori & Awajan (2021)* only reviewed the contributions of those Arabic QA studies without providing adequate details on the recent DL approach and models. Moreover, a majority of the survey and review papers did not reviewing existing Arabic QA datasets. *Biltawi, Tedmori & Awajan (2021)* covered six Arabic QA datasets, which was not sufficient for detailed analyses. A majority of the survey and review papers listed in Table 2 do not focus on DL as a recent approach to Arabic QA systems. Additionally, no study presented Arabic QA taxonomy that would assist researchers in the field of Arabic QA systems. Conversely, no studies have focused on the challenges of current Arabic QA systems in recent years nor have proposed a solution to improve the accuracy of existing Arabic QA systems and models.

This research article provides a SLR of existing Arabic QA studies to address the restrictions of previous reviews on Arabic QA tasks. To improve search results, a large number of bibliographic databases were used along with a broader range of keywords that consider various naming conventions of QA tasks. Therefore, 40 primary studies related to Arabic QA were selected. Moreover, the selected studies in the QA-proposed domain, their strengths and weaknesses, proposed system/model approaches, datasets used, methodologies, and evaluation results were thoroughly analyzed. The analysis results assisted in the identification of issues and limitations existing in the current Arabic QA literature, and potential avenues for future research were identified to motivate participation of researchers in Arabic QA tasks.

## SURVEY METHODOLOGY

SLR is a research process that gathers, evaluates, and analyzes important published papers in order to precisely answer research-related questions concerning a certain issue. The

Alrayzah et al. (2023), *PeerJ Comput. Sci.*, DOI 10.7717/peerj-cs.1633

**Table 2** Existing related works.

| Ref. | Review type | Purpose | No. of studies | Years of studies | Arabic QA taxonomy? | No. of Dataset | DL approach/ models? | Evaluation metrices? | Limitations |
|---|---|---|---|---|---|---|---|---|---|
| Ezzeldin & Shaheen (2012) | Survey | -Reviewing challenges of Arabic language in QA task. -Reviewing three main tasks of QA: question analysis, passage retrieval and answer extraction. - Summarizing main Arabic QA tools. | N/A | N/A | N/A | N/A | N/A | N/A | Paper did not present studies for Arabic QA task |
| Ray & Shaalan (2016) | Review | -Presenting Arabic QA techniques, tools, and computational linguistic resources. -Reviewing challenges of Arabic language in QA task. -Reviewing three main tasks of QA: question analysis, passage retrieval and answer extraction. | N/A | N/A | N/A | N/A | N/A | N/A | Paper did not present studies for Arabic QA task |
| Bakari, Bellot & Neji (2016a) | Literature Review | -Review of the main approaches and experimentations in Arabic QA systems. -Comparing existing Arabic QA systems. | 11 | 1993–2014 | N/A | N/A | N/A | N/A | Studies till 2014 |
| Bakari, Bellot & Neji (2016b) | Review | -Reviewing the main approaches of Arabic QA. -Discussing the different proposed systems with a classification. | N/A | N/A | N/A | N/A | N/A | N/A | Paper did not present studies for Arabic QA task |
| Utomo, Suryana & Azmi (2020) | SLR | -Disusing research issues, morphology analysis, question classification, and ontology resources of Holy Qur'an QA systems. -Reviewing and comparing existing studies for Holy Qur'an QA systems. | 9 | 2008–2016 | N/A | N/A | N/A | N/A | Review paper only for studies built for Holy Qur'an QA task. Also, Paper did not present any review about DL approach and recent models. |
| Alwaneen et al. (2021) | Survey | -Reviewing challenges of Arabic language in QA task. -Presenting Arabic QA techniques, tools, and evaluation metrics. -Reviewing and comparing existing Arabic QA systems. | 22 | 2002–2020 | N/A | N/A | N/A | Yes | Paper did not present any review about DL approach and recent models. |
| Biltawi, Tedmori & Awajan (2021) | SLR | -Reviewing challenges of Arabic language in QA task. -Presenting Arabic QA techniques, tools, and evaluation metrics. -Presenting some of Arabic QA datasets. -Reviewing and comparing existing Arabic QA systems. | 26 | 2002–2020 | N/A | 6 | N/A | Yes | Paper did not present any review about DL approach and recent models. |
| Mahdi (2021) | Survey | -Reviewing and comparing existing studies of QA systems which are based on BERT. | 9 | 2019–2020 | N/A | N/A | N/A | N/A | Review paper only for studies built based on BERT model for QA task. Also, Paper did not present any review about DL approach and recent models. |
| Present study (2023) | SLR | -Reviewing challenges of Arabic language in QA task. -Reviewing approaches of Arabic QA systems. -Presenting some of Arabic QA datasets. -Reviewing and comparing existing Arabic QA systems. -Including the first Arabic QA taxonomy for techniques, domains, approaches, datasets, and components of existing systems. -Discussing recent trends of Arabic QA systems. -Presenting limitations of existing Arabic QA systems. -Proposing future directions for research in Arabic QA. | 40 | 2013–2022 | Yes | 21 | Yes | Yes | There is no Arabic QA studies founded in 2023. |

**Notes.**

Ref, reference; N/A, not available; SLR, systematic literature review.

major objective of an SLR is to provide a method for searching present literature that is repeatable, unbiased, and exhaustive. The SLR conducted for this study followed the guidelines developed by *Kitchenham & Charters (2007)*. The guidelines can be summarized into three main phases: review planning, review conduction, and writing/publishing the review findings. Review planning process involves defining the reason for the review, establishing research questions, and developing the review methodology. A review protocol is a document that specifies all the phases in the SLR process, from choosing primary studies to collecting and synthesizing data from those studies. The second phase involves review conduction, where the steps outlined in the protocol are followed. The final phase includes writing and publishing the review results, validating the findings, and ensuring accurate and well-supported results.

## Objective

In 'Survey methodology', existing Arabic QA-related review papers requiring an adequate coverage on Arabic QA tasks have been discussed. This study aims to systematically review existing studies on several Arabic QA systems/models to identify gaps and areas for future research.

## Research questions

Multiple factors can affect the performance and robustness of Arabic QA systems. These factors include utilized datasets, information retrievers, questions and documents analyses, answer extracting algorithms, approaches used, and evaluation methods used to assess the suggested solutions (*Calijorne Soares & Parreiras, 2020*; *Cambazoglu et al., 2020*; *Elfadil, Jarajreh & Algarni, 2021*).

Considering the aforementioned factors, research questions (RQs) were designed, which are as follows:

RQ1: Which are the Arabic QA studies conducted to date?

RQ2: Which are the Arabic QA datasets used to evaluate the Arabic QA systems and models proposed by the researchers, and which datasets are available to the public?

RQ3: Which evaluation criteria were utilized to assess the Arabic QA systems and models?

RQ4: What are the current research methods and status of Arabic QA models and systems?

RQ5: What are the limitations of Arabic QA studies?

RQ6: How to enhance the Arabic QA systems and models?

## Search procedure
### Study resources

The automated Google Scholar academic search engine was used to search for relevant and published studies. The publication period of the primary studies was set between 2013 and 2022 to include recent Arabic QA studies within the last decade.

### Search keywords

Since different Arabic QA techniques and resources use a variety of terminologies, generic and common search terms were used to obtain a broad overview of the field.

**Table 3  Criteria for inclusion and exclusion.**

| Inclusion criteria |
|---|
| Study is written in English and published sometime between 2013 and 2022. |
| Language of the dataset is MSA or Arabic. |
| Type of the dataset used in the study is factoid QA dataset, starting with a wh- question. |
| Study handles Arabic QA tasks. |
| Research findings are made available on the website of an academic institution or through publication in a journal or conference by undergoing peer-review. Additionally, the results are included in doctoral and master's theses and dissertations. |

| Exclusion criteria |
|---|
| Proposal studies that are not empirically tested. |
| Dataset is in English, non-Arabic, Arabic dialectics, or community QA, with |
| classifying questions, generated questions, Yes/No QA, Multiple-Choice QA. |
| Study handles chatbot as Arabic QA. |

Arabic QA studies were collected primarily through Google Scholar by using different search keywords, such as "question-answering systems," "question answering," "question answering models," "Arabic question answering systems," "Arabic QA models," "Arabic question answering tasks," "Arabic question answering dataset," "Arabic QA dataset," "question answering dataset," and "question-answering."

### Search string

To represent all possible root-word endings, a wildcard (*) was used; for example, answer* represents both "answering" and "answers." Furthermore, Boolean operators (AND and OR) were used to account for synonyms, spelling varieties, and naming inconsistencies. In some cases, synonymous terms in Google Scholar were separated into search strings because using them with the OR operator yielded hundreds of results that were irrelevant. The following search strings were used for automatic searches in Google Scholar.

- ("question answering models" OR "question answering systems") AND ("Arabic")
- ("question answer*") AND ("Arabic")
- ("QA") AND ("Arabic")
- ("Question-Answer*") AND ("Arabic")
- ("question answering dataset") AND ("Arabic")
- ("QA dataset") AND ("Arabic")
- ("QA dataset") AND ("question answer*") OR ("Arabic")
- ("question answering dataset") OR ("question answer*") AND ("Arabic")

### Study selection

The collected studies had to meet specific selection criteria before being considered in this review. A set of inclusion and exclusion criteria was established to ascertain the eligibility of the studies, as shown in Table 3. The studies that met the inclusion criteria were part of the review and analysis, while those that met at least one of the exclusion criteria was excluded.

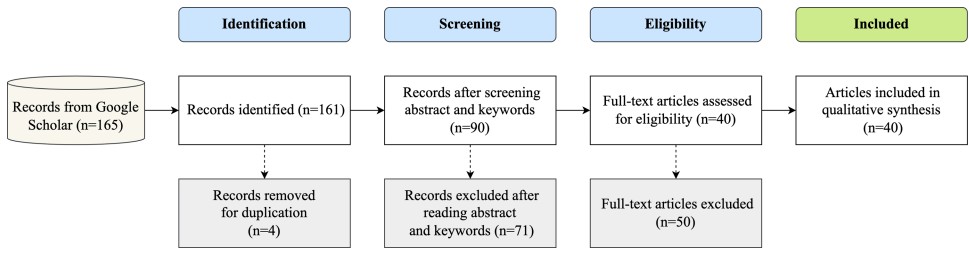

**Figure 2** **Flow of search and selection process using PRISMA technique.**

### Search process

A preliminary search on Google Scholar was conducted to collect publications within a search period between 2013 and 2022 to ensure the coverage of the most significant primary research and include early works in Arabic QA. During this stage, relevant primary studies were identified by conducting a literature survey using pre-determined criteria of inclusion and exclusion and strings of the search. The search procedure began with a title screening, and publications with titles meeting one of the selection criteria were collected. When the study title contained the phrase "question-answer" or one of its synonyms, in addition to the keyword "Arabic," the study was included. However, when more than one search string was used, some duplicate publications were found. Hence, duplicates were removed from the list of candidate research after screening their titles. Thereafter, the abstracts and keywords of the published studies were screened to meet the inclusion and exclusion criteria. Next, the full text of the collected publications were scanned when the abstract and keywords were not highly conclusive. The total number of results provided by Google Scholar is shown in Fig. 2, including the scanning step results obtained using the preferred reporting items for systematic reviews and meta-analyses (PRISMA) technique.

### Data extraction

The instructions of *Kitchenham & Charters (2007)* were applied for data extraction. Information including details on publisher, authors, publication date, and publication type was extracted from the primary studies that were included in the analyses. Additionally, data relevant for answering research questions were collected, such as domains, methodologies, text and question analysis, ways of extracting answers, datasets, and reported results. The gathered data were organized in a table for analyses.

## RESULTS

The findings of the SLR conducted for this study are discussed in this section, with an overview of the included studies. Analytical results of the data extracted from the publications included in this SLR are also discussed to answer the pre-determined research questions.

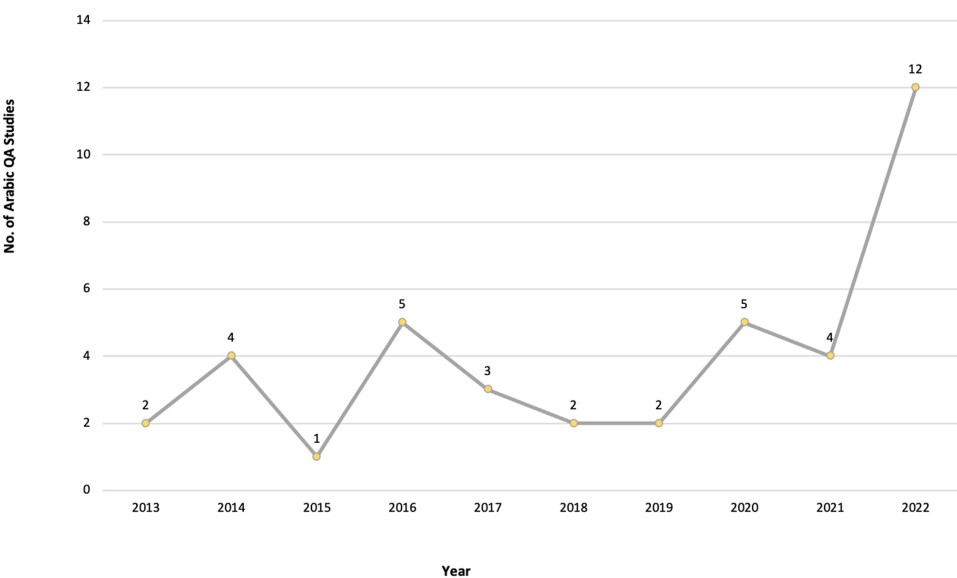

**Figure 3** **Number of Arabic QA studies conducted between 2013 and 2022.**

## Summary of the included studies

After meeting the inclusion and exclusion criteria, 40 primary studies were selected for this review, as illustrated in Figs. 3, 4 and 5, which depict the distribution of the included studies based on the publication year, type, and publisher, respectively. The primary studies included were published between 2013 and 2022, with a majority published in 2022. The studies were published by a variety of publishers, the most common being computational linguistics journal (ACL) and arXiv that offer free digital archive of papers in NLP and computational linguistics from conferences and journals. A majority of the publications were journal articles, followed by conference proceedings.

## RQ1: Which are the Arabic QA studies conducted to date?

Based on the inclusion and exclusion criteria for selecting primary Arabic QA studies, the selected studies are presented and analyzed in this section to answer RQ1.

*Ezzeldin, Kholief & El-Sonbaty (2013)* suggested ALQASIM as an Arabic QA system for analyzing documents related to posed questions based on several NLP tools. They used morphological analysis and disambiguation for Arabic (MADA), TOKAN morphological analyzer, part-of-speech (POS), Arabic WordNet (AWN), and stop words removal techniques. MADA extracts morphological and contextual information from unprocessed Arabic texts, while the process of POS involves assigning a word to a specific part of speech, which includes nouns, punctuation, verbs, adjectives, adverbs, conjunctions, pronouns, or prepositions. AWN is a lexical database that mirrors the developmental process of Princeton University developed English WordNet and Euro WordNet, but for Arabic. It contains various lexical and semantic relations between word senses. For extracting the correct answer, the distance between the question and answer choice locations was subtracted from the sum of the scores at each location, and the resulting score was used to

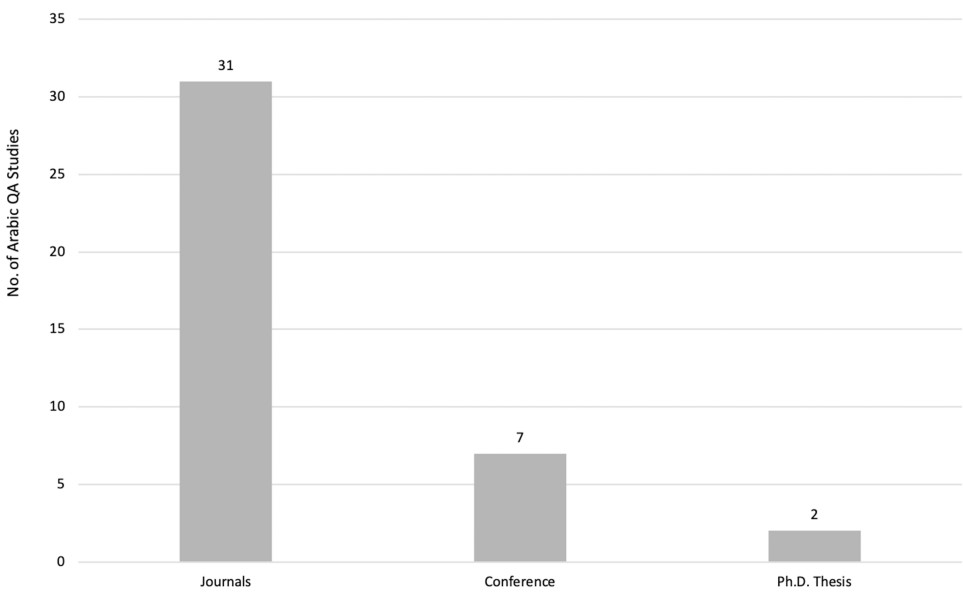

**Figure 4** Distribution of Arabic QA studies based on publication type.

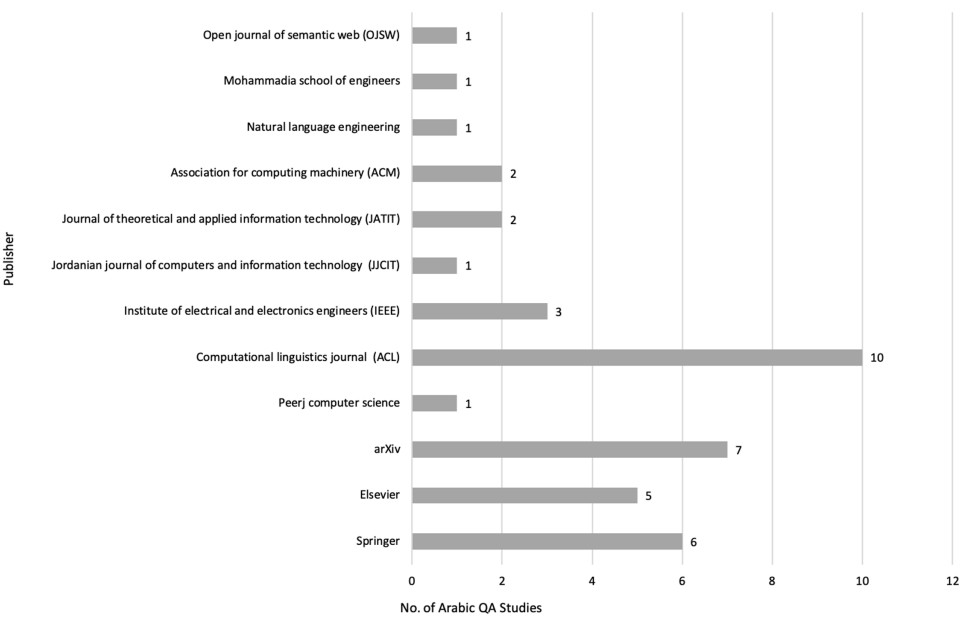

**Figure 5** Distribution of Arabic QA studies based on publishers.

determine the location of the correct answer. This ensured that the answer locations with the highest scores were those that were closest to the question location.

*Fareed, Mousa & Elsisi (2013)* proposed a retrieval Arabic QA system based on three modules. The first module involved question analysis using AWN, which built "synonym, definition, subtype, and super type" relations for each keyword in the question. The second module was a passage retrieval module that used the Google search engine, java IR system (JIRS), and Khojah stemmer. Google was used to acquire documents related to the question, while JIRS was used to re-rank the Google search results on a structural basis to improve their ranking. The Khojah stemmer, an Arabic stemming algorithm developed by Shereen Khoja (*Fareed, Mousa & Elsisi, 2013*), is one of the most popular light stemmers for Arabic language. It was use to strip words of their longest prefix and suffix. The words were then checked against verbal and noun patterns to find a match. The third module involved answer extraction based on the type of the question.

*Abdelnasser et al. (2014)* suggested the use of ML algorithms and NLP tools for building a QA system on Islamic data, especially for the Holy Qur'an to explain its verses. They exploited an ML algorithm known as support vector machine (SVM) to classify questions after analyzing them using an Arabic NLP toolkit or MADA. MADA extracts morphological and contextual information from unprocessed Arabic texts, which are then used for more accurate POS tagging, lemmatization, diacritization, and stemming.

*Kamal, Azim & Mahmoud (2014)* applied for Fatwa in Islamic data by proposing the use of latent semantic indexing (LSI) for indexing and retrieving documents automatically. LSI uses singular value decomposition (SVD) as a mathematical method to identify patterns in relationships between words and concepts in raw texts. For ranking candidate answers, a term document matrix (TDM) was used by applying term frequency-inverse dense frequency (TF-IDF) algorithm for each word in the query using a math equation. The algorithm used the frequency of words to determine how relevant those words were to a given document. This was done by observing the appearance frequency of words in the document compared to their appearance frequency in other documents.

*Sadek (2014)* enhanced Arabic "why" and "how to" questions by applying text mining methods. The author combined pattern recognizer model with text parser model to form a single system. The pattern recognizer model used linguistic patterns to identify relationships present in sentences, with approximately 900 patterns available in the system. The model was capable of extracting information related to cause–effect and method–effect. Thereafter, the text parser model was built based on the framework of rhetorical structure theory (RST) of text structure, which was used to assist in the creation of computational text plans. The text structure in RST is based on a hierarchy of small patterns, called schemas. The RST approach recognizes a text as a group of clauses arranged in a hierarchical structure and interconnected in different ways, rather than being a mere sequence of clauses. A similar theoretical approach was used in a research by *Sadek & Meziane (2016)* for a similar purpose but with different datasets.

*Abouenour (2014)* proposed an Arabic QA system called IDRAAQ, which combined three different approaches based on levels, such as keyword, structure, and semantics. The three approaches were QE process based on AWN semantic relations, N-gram model

for passage retrieving, and conceptual graphs (CGs) for extracting answers. The N-gram model referred to a consecutive series of N words extracted from a particular text, which could be utilized to estimate the likelihood of the subsequent word in the sequence based on the preceding word. The CGs presented graphical representations of the relationships between words and concepts. They were composed of nodes and edges, which represented the concepts and the relationships between them, respectively.

*Ahmed & Anto (2016)* aimed to analyze open-domain questions to extract correct answers. For analyzing questions, several steps were proposed, such as tokenization, stop word removal, POS tagger, and classifying question using the SVM algorithm. Thereafter, a vector space model (VSM) was used to retrieve documents, and then extract answers. Before tokenizing the questions, an Arabic tree bank (ATB) was used for morphological analyses. It was a corpora annotated for morphological information and POS with syntactic properties.

A framework known as semantic question answering (SQA) was proposed by *Tatu et al. (2016)* for large document collections. The aim of the framework was to provide an opportunity for users to retrieve stored information using natural language questions. The questions were automatically transformed into SPARQL queries and utilized to investigate the semantic index of the resource description framework (RDF). SPARQL is a protocol and semantic query language for RDF, which is designed to retrieve and manipulate data stored in the RDF format. The RDF involves statements used for describing and representing data in a structured manner. It consists of a set of rules for representing data in the form of triples, composed of a subject, predicate, and an object. The RDF allows organization of data into a graph structure, which can be used for querying and interpreting data.

*Shaker et al. (2016)* applied an ontological approach on an Islamic dataset called Fatwa. The study aimed to propose a QA approach for the domain of Islamic Fatwa utilizing ontology-based techniques. Hence, an ontological model was constructed through a compilation of Fatwas from ibn uthaymeen-prayer Fatwas.

*Al-Chalabi, Ray & Shaalan (2015)* applied a semantic approach to search and retrieve documents before answer extraction. The authors used a query expansion (QE) method to add semantically equivalent terms to increase the likelihood of retrieving documents containing relevant information. QE is a technique employed in IR to reduce query–document mismatch and boost retrieval performance. This was implemented by choosing and incorporating terms into the query of the user.

A hybrid approach combining RB and ML methods was used by *AlShawakfa (2016)*. The author built a PoS tagger with combination of RB approach. The tagger employed over 35 tagging rules to recognize various types of nouns, including "Proper, Action, Genus, Agent, and Patient nouns," along with adjectives, adverbs, demonstrative nouns, instrument, time, and place (*AlShawakfa, 2016*). Additionally, more than 15 and 17 tagging rules were used to identify particles and verbs, respectively. For IR, the relational database system employed VSM for locating and fetching relevant documents. The mathematical model entailed representation of each document and query as a vector and involved assigning weights to the index terms in both entities. Similarity between the query and each document was determined based on these weights, which allowed the model to identify the

documents with highest relevancy to the query. Therefore, the best results were retrieved. Cosine similarity is a measure of how similar two vectors are to each other, and is often used to compare the similarity of documents. Through the analysis of cosine similarity between a query and a set of pertinent documents, the best possible answers for the query can be accurately determined.

Another system known as LEMAZA was proposed for Arabic "why" questions by *Azmi & Alshenaifi (2017)*. They applied the RB approach to RST and aimed for IR in open-domain QA field. The documents were processed by breaking them into tokens, normalizing them, removing stop-words, and stemming.

*Nabil et al. (2017)* proposed a system to improve the process of retrieving passages from numerous documents by analyzing the question, retrieving the document, and then extracting the answer. This approach is used most commonly by Arabic QA systems. QE with SVM algorithm was used to analyze and classify questions. An improved version of MADA, known as MADAMIRA, for Arabic text was also used. MADAMIRA ran faster than MADA with more analyzing rules.

*Albarghothi, Khater & Shaalan (2017)* applied semantic web and ontological technologies to enhance QA tasks in the pathological domain. The ontology was built using a Protégé tool, which translated inquiries into triple patterns and built SPARQL queries to access RDF data. Protégé is a widely used tool for representing and reasoning knowledgeable concepts and defining the properties and instances of those concepts. According to the ontological dictionary, Protégé is defined as a RDF/web ontology language (OWL) file. OWL provided a wide range of constructs for representing the semantics of a domain and allowed for the expression of complex ontological relationships. Furthermore, it provided support for reasoning and enabled applications to infer new knowledge from the ontology. Thereafter, Jena framework was used for answer extraction and to build applications that made use of semantic web and linked data technologies. It allowed users to easily extract data from and write to RDF graphs.

*Ben-Sghaier, Bakari & Neji (2018)* built a system known as NArQAS based on the RB approach. Recognizing textual entailment (RTE) technique was used to identify the correct answer from a set of candidates. RTE was combined with several operations achieved by NLP tools, such as IR, information extraction, automatic language processing, and automatic reasoning.

*Ismail & Homsi (2018)* proposed a dataset of Arabic "why" questions. The authors calculated the probabilities of rhetorical relations (RR) to extract relevant answers for the posed questions. The RR indicated the logical relationship between two sections of texts. Ismail and Homsi (*Ismail & Homsi, 2018*) used RR to determine related reasons and causality in texts. Moreover, ML algorithms, such as SVM, were used to classify the selected documents from the internet into eight domains. NLP techniques such as bag-of-words were used to convert documents into vectors of weighted frequency for each token before RR was calculated. For retrieving answers, TF-IDF was used. However, the study was only focused classification and analysis of questions, instead of extracting correct answers.

To improve IR using a transformer model, *Mozannar et al. (2019)* built a retrieval system using TF-IDF. The system retrieved those Arabic text from Wikipedia that were related

to user-generated questions, before passing the most relevant documents to the readers. Bidirectional encoder representation (BERT) was used as a reader in this study.

*Al-Shenak, Nahar & Halawani (2019)* used ML algorithms such as SVM, SVD, and LSI to classify questions. Based on these classifications, relevant documents containing answers were retrieved.

*Bakari & Neji (2020)* aimed to design a system to retrieve text from the internet. The authors converted both the questions and passages into semantic and logical representations. The questions and passages were first converted to CGs, which were then converted to logical representations. Textual entailment relations among the logical representations were extracted, and the most relevant passages were determined based on those relations.

Several pre-trained transformer models used to improve the accuracy of QA systems are available. These include multilingual BERT (mBERT) (*Mozannar et al., 2019*), AraBERT (*Antoun, Baly & Hajj, 2020a*), ARAGPT2 (*Antoun, Baly & Hajj, 2020c*), and ARAELECTRA (*Antoun, Baly & Hajj, 2020b*). These models were built to improve the understanding of text by computational systems and are used for QA and named entity recognition and sentiment analysis.

IR can be further improved with graph ontology, as demonstrated by *Zeid, Belal & El-Sonbaty (2020)*. The study applied semantic operations to questions to expand the queries. Thereafter, the queries were used to search for data using graph ontology. This model followed three steps: question processing, document processing, and answer extraction. Each step was achieved using traditional NLP techniques.

*Alamir et al. (2021)* propose another system for open-domain QA tasks. The system consisted of three main stages: preparing data, processing data, and extracting answers. The TF-IDF and cosine similarity algorithms were used to retrieve the documents. For processing, a few NLP techniques were used. The dataset was built under the "Ministry of Human Resources and Social Development in the Kingdom of Saudi Arabia." However, the system handled only those words that were similar in both the question and the document. Further improvements to the system were proposed to also handle the semantics of words for more accurate results.

*Maraoui, Haddar & Romary (2021)* aimed to create an Arabic QA system specialized in answering factoid questions, particularly relating to Islamic sciences. To extract answers, the system consisted of three phases: analyzing the question, searching for information, and processing the answer. The first stage of QA involved analysis of the question in order to formulate an appropriate query. After selecting a specific set of elements from a database, the second stage involved IR through an information search. During the third stage, the answer processing phase provided an accurate Arabic response. The system was built using a normalized database that adhered to the text encoding initiative (TEI) standard. TEI is a consortium of scholars and researchers who work together to develop and maintain a standard for representing texts in digital form.

Two deep bi-directional transformers for Arabic known as ARBERT and MARBERT were proposed by *Abdul-Mageed, Elmadany & Nagoudi (2021)*. Both transformer models were built and implemented based on the BERT transformer model. The models differed

in their training data as some of the models trained on MSA, social media data, or Arabic dialects. However, the models had the same stages as that of the BERT model. Additionally, the study proposed a new large benchmark known as Arabic natural language understanding evaluation benchmark (ARLUE). The ARLUE composed of 42 Arabic datasets for Arabic NLP tasks. However, the result of fine-tuning the model on Arabic QA task was not competitive because some questions were based on Wikipedia articles, and the two models were not trained on Arabic Wikipedia articles.

To build and train Arabic models based on text-to-text transfer (T5) (*Raffel et al., 2020*) and mT5 (*Xue et al., 2021*) transformer models, *Nagoudi, Elmadany & Abdul-Mageed (2022)* proposed and implement AraT5 for several tasks. T5 and BERT are different in that T5 has a casual decoder and uses the task of fill-in-the-blank cloze instead of a masking task. The AraT5 had a task of question generation under the mode of read the passage and generate question. A new Arabic QA dataset named ARGEN $_{QG}$ was developed. The dataset was created by gathering sets of QA dataset that included Arabic reading comprehension dataset (ARCD), multi-lingual QA (MLQA), cross-lingual question answering dataset (XQuAD), and typologically diverse languages question answering (TyDiQA). However, the accuracy for QA was unsatisfactory, and the reasons for low accuracy were excluded for this study.

*Alsubhi, Jamal & Alhothali (2022)* suggested improvements to IR modules by applying dense passage retrieval (DPR) techniques instead of TF-IDF and best match 25 (BM25). The suggested solution was intended to develop open-domain QA systems. The proposed DPR method was used to read and rank passages with accurate results within shorter periods than ML methods. Being a retrieval-based method, the DPR method used a dense vector representation of text to identify relevant passages from a collection of documents in open-domain QA systems. A transformer model was utilized to encode passages, and then a scoring function was used to rank the passages according to their relevance to a given query (*Alsubhi, Jamal & Alhothali, 2022*). BM25 served as a ranking mechanism employed by search engines to assess the relevancy of a document to a specific search query. It was built based on SVM, but with addition functions of IR tasks, such as document classification and clustering. Additionally, BM25 finds applications in text vectorization, which is a process of transforming text into numerical vectors that can be used to determine the similarity between documents. However, the QA datasets considered by *Alsubhi, Jamal & Alhothali (2022)* were not enough for training the system because a low resource dataset or Arabic dataset was used.

As an extra step for IR, *Hamza et al. (2022)* classified questions before candidate answer extraction. They aimed to reduce searching space for correct answers by the use of three embeddings for the questions. The first embedding involved the use of word representations in BERT, embedding from language models (ELMo), AraBERT, and word to vector (W2V). Fused embedding with fine-tuning questions was the second method for embedding, while the third method involved utilizing a "boom one head" neural fusion model to derive features from both fine-tuning and existing embeddings of the questions. Additionally, the study used vanilla classifier to classify the questions. However, only the accuracy results

of the classifier methods were discussed, while its application to Arabic QA datasets were excluded.

To increase the accuracy of QA systems utilizing datasets from the Holy Qur'an as an Islamic dataset, several studies were published in 2022 in the "*5th Workshop Open-Source Arabic Corpora and Processing Tools with Shared Tasks on Qur'an QA and Fine-Grained Hate Speech Detection,*" some of which are discussed as follows.

*Premasiri et al. (2022)* proposed the use of transfer learning (ensemble learning) to improve and increase Arabic QA datasets. Contributions to the field of Arabic QA were aimed by presenting Qur'an QA datasets. Different versions of Arabic transformer models, such as AraELECTRA, camelbert, mBERT, and AraBERT, were introduced. Thereafter, self-ensemble methods were applied to predicted the most accurate answers to the questions and overcome different results. However, the study did not consider the results of fine-tuning the transformer models using the Holy Qur'an dataset.

Furthermore, *ElKomy & Sarhan (2022)* attempted to improve the accuracy of the Qur'an QA dataset. An ensemble approach with post-processing operations after the fine-tuning stage was proposed in the study, which used BERT, AraBERT, ARBERT, and MARBERT transformer models to collect all prediction result of answers that were then merged. This approach was termed as ensemble-vanilla approach. Conversely, *ElKomy & Sarhan (2022)* suggested and implemented post-processing approaches by ranking the top 20 span answers. The answers were listed by a single model of the set model used in the ensemble approach or by the span-voting ensemble approach. However, variations were observed in the reported results owing to using extensive cross-validation.

*Alnajjar & Hämäläinen (2022)* designed a model based on multilingual BERT and fine-tuned it especially for Qur'an QA dataset. They reimplemented multilingual BERT model with an addition of Islamic data for pre-training the model. The Islamic data were collected from different Islamic websites and they provided explanations of the Qur'an (Tafseer and Fatwas). Thereafter, a comparison between the designed and multi-lingual BERT models was drawn. The new model had followed the post-processing step after predicting the answer to check the answer length. For example, questions on "who" should have short answers. Conversely, the new model scored approximately 34% in partial reciprocal rank (pRR).

To overcome the overfit of training models on Qur'an QA datasets, *Aftab & Malik (2022)* suggested and implemented regularization techniques such as data augmentation and wight-decay. The BERT model was used to train models, which were then fine-tuned to Qur'anic reading comprehension dataset (QRCD). The results showed improvement after using both regularization techniques. The use of another technique such as back translation was suggested to increase the Arabic QA dataset and enhance the performance of QA.

*Mostafa & Mohamed (2022)* suggested solutions to improve the performance of Qur'an QA using QRCD. For the study, AraELECTRA was used to train the model on Arabic QA dataset to enhance the fine-tuning phase. Before fine-tuning the model on the QRCD, TyDi QA, SQuAD, and ARCD were employed. Moreover, *Mostafa & Mohamed (2022)* experimented several loss functions, such as cross-entropy, focal-loss, and dice-loss, to find

the best function for increasing the performance of their model. Among the considered loss functions, cross-entropy showed the best performance.

To develop an answer on QRCD, *Touahri et al. (2022)* designed a sequence-to-sequence model based on the mT5 language model. The model was implemented as base, large and extra-large. After training on the train set of the QRCD, the model was fine-tuned. Thereafter, the development set was used for evaluation, and the test set was utilized to generate predictions. The model had high performance on the development set. However, the performance was much lower on test set and a degradation in performance results was observed.

*Alsaleh, Althabiti & Alshammari (2022)* used three Arabic models, AraBERT, CAMeL-BERT, and ArabicBERT, for fine-tuning on QRCD. The experimental results showed that AraBERT outperformed the other Arabic pre-trained models. However, the highest result of accuracy still required improvements, especially when related to religion questions.

*Singh (2022)* suggested three solution for fine-tuning using QRCD. Three techniques known as semantic embeddings and clustering, Seq2Seq based text span extraction, and fine-tuning BERT model were used. The first technique was aimed at using sentence embedding for each answer in QA dataset using AraBERT. The second was aimed at generating the text span of answers using mT5 and mBART. Finally, the third solution involved the use of BERT. The suggested model treated a question-passage pair as a unified sequence and subsequently transformed it into an input embedding. Higher results were obtained from the third solution.

*Keleg & Magdy (2022)* implemented improvements in fine-tuning Arabic BERT on QCRD. They categorized datasets based on question types. Additionally, they built faithful splits to generate new training and development datasets. Thereafter, the datasets were concatenated to detect data leakage between two splits of the dataset. However, the results of accuracy were unsatisfactory as the size of the dataset was not enough for training.

An ArabicTransformer model was proposed by *Alrowili & Shanker (2021)*. The model comprised ELECTRA-objective and a funnel transformer. The funnel transformer was used to decrease the count of hidden states in a sequence by utilizing a pooling method, which results in a substantial decrease in the pre-training costs. To compress the complete sequence of hidden states of the encoder into a set of blocks, a pooling technique was employed. The ArabicTransformer model had one architecture B6-6-6, which consisted of three blocks. Each block had six layers, with a hidden size of 768. A B4-4-4 model design comprises three blocks, each having a hidden size of 768 across four layers. The model was trained on several tasks such as QA. However, the model was tested on only two QA datasets.

Data listed in Table 4 present a summary of the selected Arabic QA studies based on the study reference and year, name of proposed system or model, approach used in building the system or model, aimed open or closed-domain of study, utilized dataset to evaluate the proposed solution, evaluation result, and the limitation or recommended for the future work.

On analyzing the data presented in Table 4, DL was observed to be the most recent approach adopted for building Arabic QA systems and models, especially from 2020.

Alrayzah et al. (2023), *PeerJ Comput. Sci.*, DOI 10.7717/peerj-cs.1633

**Table 4   Arabic QA studies conducted between 2013 and 2022.**

| Ref. | System or model/novelty | Approach | Domain | Methodology | Dataset | Evaluation result | Limitations/Future work |
|---|---|---|---|---|---|---|---|
| *Ezzeldin, Kholief & El-Sonbaty (2013)* | ALQASIM → It introduced a novel technique for analyzing reading test documents instead of questions for answer selection and validation. | RB | Open-domain | Analyze the reading documents instead of the questions using the following NLP tools: MADA and TOKAN morphological analyzer, POS, AWN, and removing stop words. To extract answers, score distance between answer and question locations. | QA4MRE, Cross-language education and function (CLEF) | Accuracy: 0.31% C@1: 36% | Low accuracy results not discussed. |
| *Fareed, Mousa & Elsisi (2013)* | N/A → It proposed a design for an Arabic question answering system based on query expansion ontology and an Arabic stemmer. | RB | Open-domain | Three main steps: Question analysis using AWN, document retrieval using JIRS, and Khojah stemmer. Answer extracting. | CLEF, text retrieval conference (TREC) | Accuracy: 38.77% Mean Reciprocal Ratio (MRR): 16.20% Answered questions: 65.55% | Authors are recommended to test the system on a larger dataset of questions. |
| *Abdelnasser et al. (2014)* | Al-Bayan → The system is specialized for the Holy Qur'an. The system retrieves the most relevant Qur'an verses and extracts the passage that contains the answer from the Qur'an and its interpretation books (Tafseer). | ML → SVM with NLP tools such as MADA | Closed-domain | Three components: Semantic IR to retrieve verses for user questions. MADA to analyze questions and SVM to classify questions. A third component to extract the ranked answers with their interpretations. | Quranic Ontology and Tafseer Books → A total of 230 questions, comprising randomly collected queries from forums and common Quranic topics, were segregated into two sets. The first set comprised of 180 questions for training purposes, while the second set consisted of 50 questions for testing. | Accuracy: 85% Precision: 73% C@1:11% | Authors are recommended to apply the proposed solution on list-type questions. |
| *Kamal, Azim & Mahmoud (2014)* | N/A → The proposed system uses information retrieval approaches to get to the closest answers to the input question, even if the question differs from the stored questions. | RB | Closed-domain | Three main steps: Question analysis → tokenization, normalization, and classifying questions. Candidate answer retrieval → NER and TDM Answer ranking → TF-IDF and SVD | Fatwa → A compilation of 3,000 distinct passages on Islamic QA, sourced from various websites, was manually assembled into a fatwa. | Recall: 95.3% MRR: 0.916 | Several processing steps for each question, document, and passage may cause delay in retrieving answers. The study is focused on IR rather than RC. |
| *Sadek (2014)* | N/A → The paper proposed a new strategy for developing QA systems for the Arabic language, specifically for answering "why" and "how to" questions. | RB | Closed-domain → Extracting answers by applying text mining approach. | Combine two models: Pattern recognizer to apply linguistic patterns and relationships among sentences. Text parser that uses RST to analyze texts from a discourse perspective. Answer extraction by VSM. | Dataset for "why" and "how to." Set of articles collected from the contemporary Arabic corpus (415 texts, 70 "why" questions, 20 "how to" questions) with their answers. | Recall: 81% Precisions: 78% Accuracy: 68% | The proposed solution was built by calculating the pattern similarity between words, which may hold different meanings. |
| *Abouenour (2014)* | IDRAAQ → It consisted of three-levels approaches to improve a system for Arabic QA task using existing resources and several techniques. | RB | Open-domain | Three-level approach based on the level of keywords, structure, and semantic meaning. Apply QE using AWN for question analysis. Use N-gram model for document retrieval. Use CGs to represent meanings between questions and passages to predict the answers. | TREC, CLEF | TREC+ CLEF: 26.76% accuracy, 11.58 MRR | The system requires larger datasets to introduce more accurate results. |
| *Al-Chalabi, Ray & Shaalan (2015)* | N/A → The proposed method to add semantically equivalent keywords in Arabic questions by using semantic resources, which can improve the accuracy of Arabic QA systems. | RB | Open-domain | Apply QE method through the addition of semantically equivalent terms to increase the likelihood of retrieving documents containing relevant information. Use AWN tool as a semantic resource to find synonyms for words in questions. | TREC, CLEF, Arabic questions. | MRR: 2.18 out of 3. | Results are not discussed in detail and are poorly presented. |

Alrayzah et al. (2023), *PeerJ Comput. Sci.*, DOI 10.7717/peerj-cs.1633

**Table 4** (*continued*)

| Ref. | System or model/novelty | Approach | Domain | Methodology | Dataset | Evaluation result | Limitations/Future work |
|------|------------------------|----------|--------|-------------|---------|-------------------|------------------------|
| *AlShawakfa (2016)* | N/A → The originality of this research lies in the creation of an extensive collection of over 60 tagging rules, 15+ question analysis rules, and 20+ question patterns. to enhance the precision and correctness of answers generated in the context of Arabic QA. | RB | Open-domain | More than 95 tags rules, question analysis rules, and question patterns are combined and used to build the system. VSM is used for IR. | Dataset collected from Wikipedia dated 2010. Contains 75 documents with 335 questions. | F1: 87% Accuracy: 78% Recall: 97% | The system is based on syntactic analysis of words, which is a less accurate and time-consuming approach. |
| *Sadek & Meziane (2016)* | N/A → The paper developed of a new Arabic text parser that is oriented towards QA systems dealing with "why" and "how to" questions. | RB | Closed-domain | Apply Arabic text parser designed for QA systems that handle "why" and "how to" questions. RST is used for describing relations in text. | "Why" and "how to" QA dataset consisting of documents from open-source Arabic corpora. | Recall: 68% MRR: 0.62. | Authors are recommended to use investigate query expansion techniques for better results. |
| *Ahmed & Anto (2016)* | N/A → The novelty of this paper lies in the use of several techniques to analyze the question, including a Stanford POS Tagger & parser for Arabic language, NER, tokenizer, Stop-word removal, question expansion, question classification, and question focus extraction components. | Hybrid approach (RB+ML) →SVM classifier | Open-domain | Analyze questions by: ATB morphological analyses, and then tokenize questions. Remove stop words. QE using AWN. Classify questions using SVM. Focus on questions using Stanford POS tagger for Arabic. retrieving document using VSM. Extract questions based on Arabic | TREC | MRR: 65% | Answer extraction based on an Arabic tagger may reduce accuracy. |
| *Tatu et al. (2016)* | SAQ → This framework transforms the semantic knowledge extracted from natural language texts into a language-agnostic RDF representation and indexes it into a scalable triplestore. | RB based on ontology | Open-domain | Transfer unstructured text in questions into RDF as structured format, and then to SPARQL to obtain the answers. | TREC | MRR: 65.82% | Dataset is not large enough to evaluate the system. |
| *Shaker et al. (2016)* | N/A → The paper introduced ontology-based approach for Arabic QA in the domain of Islamic Fatwa. | RB based on ontology | Closed-domain | Comprises several components including: Question pre-processing → stop word removal. Question analysis → similarity of cosine and Jaccard algorithms to classify questions. Question expansion → words in the query are analyzed in terms of semantic, morphology, and spilling errors. Specific-domain ontology → constructed using TFIDF from the dataset. Open-domain ontology → constructed using AWN. | Fatwas collected from Ibn-Othaimeen Prayer Fatawas Book. | Closed-domain: precision: 72% Recall: 59% F1: 65% Open-domain: precision: 92% Recall: 90% F1: 91% | Authors are recommended to extend the classes of ontology to include more Islamic concepts |

Alrayzah et al. (2023), *PeerJ Comput. Sci.*, DOI 10.7717/peerj-cs.1633

**Table 4** (*continued*)

| Ref. | System or model/novelty | Approach | Domain | Methodology | Dataset | Evaluation result | Limitations/Future work |
|---|---|---|---|---|---|---|---|
| *Azmi & Alshenaifi (2017)* | Lemaza → It is an Arabic why-question answering system that uses the RST to automatically answer why-questions for Arabic texts. | RB | Open-domain | Utilize RST. The process of analyzing question, pre-processing and retrieving document, and extracting answer are broken down into four components. | "Why" QA dataset consisting of documents from open-source Arabic corpora with 110 "why" question and an- | Recall: 72.7% Precisions: 78.7% C@1: 78.68% | Authors are suggested to expand the test collection by incorporating a more extensive corpus, resulting in an increase in the number of |
| *Nabil et al. (2017)* | AlQuAnS → It introduced a new answer extraction pattern that matches the patterns formed according to the question type with the sentences in the retrieved passages in order to provide the correct answer. | Hybrid approach (RB+ ML) → SVM classifier | Open-domain | IR using semantic QE and ranking retrieved passages using a semantic-based process called MADAMIRA. | CLEF, TREC | Accuracy = 15.30% | Researchers are suggested to use DL techniques in future works. |
| *Albarghothi, Khater & Shaalan (2017)* | N/A → The paper introduced system based on ontology that utilizes semantic web and ontology technologies to represent domain-specific data that can be used to answer natural language inquiries. | RB based on ontology | Closed-domain | Build pathology ontology → using semantic web and Protégé tool Select questions and process it. Answer questions → using Jena framework based SPARQL query. | Pathology dataset | Precession: 81% Recall: 93% F1: 86% | Authors should cover more domains. |
| *Ben-Sghaier, Bakari & Neji (2018)* | NArQAS → The system combines reasoning procedures, NLP techniques, and recognizing textual entailment technology to develop precise answers to natural language questions. | RB → combination of semantic analyzer with logical reasoning; | Open-domain | Three main steps: question analysis, passage retrieval, and logical representation of relationships in text. RTE technique used to extract the exact answer among several candidates. | A collection of 250 questions gathered from TREC, CLEF, FAQ, and online forums forming a corpus of question-based texts. | Accuracy: 68% for answering factoid questions from the web. | Use and integration of NLP tools with semantic analyzer is time-consuming. Authors are suggested to use DL word embedding. |
| *Ismail & Homsi (2018)* | N/A → The paper introduced a new publicly available dataset called DAWQAS, which consists of 3205 why QA pairs in Arabic language. | Hybrid approach (RB+ ML) → SVM, and bag of | Closed-domain for "why" questions | SVM classifies selected documents from the web. Bag-of-words convert documents into vectors. RR is calculated. Retrieve answers by TF-IDF. | DAWQAS | F1 = 71% | Authors are suggested to add more examples to train the algorithm using DL and classify the questions automatically. |
| *Mozannar et al. (2019)* | SOQAL → The paper introduced ARCD dataset. Also, it used TF-IDF approach for document retrieval and BERT for neural reading comprehension. | Hybrid approach (RB+ DL) → TF-IDF and BERT | Open-domain | Retrieve Arabic documents related to a question using TF-IDF; pass those documents to BERT to extract an answer. | Arabic-SQuAD, ARCD | F1-Arabic-SQuAD = 48% EM-Arabic-SQuAD = 34% F1-ARCD = 51% EM-ARC = 19% | The study uses BERT as a reader, which is not a model specific for Arabic text such as AraBERT. |
| *Al-Shenak, Nahar & Halawani (2019)* | AQAS → The novelty of this paper is proposing an enhanced method and system for Arabic QA. The proposed system uses SVM, SVD, and LSI to classify the query in two phases. | ML → SVM, SVD, and LSI | Open-domain | SVM, SVD, and LSI are used for classifying the questions and retrieving relevant information. | TREC | Results for classification step: Precision = 98% Recall = 97% F1= 98% For querying: Precision (average): 88% Precision (minimum): 17% | The study focuses on classifying questions rather than extracting correct answers. |

| Ref. | System or model/novelty | Approach | Domain | Methodology | Dataset | Evaluation result | Limitations/Future work |
|------|------------------------|----------|--------|-------------|---------|-------------------|-------------------------|
| *Bakari & Neji (2020)* | NArQAS → The system integrates RTE technique with semantic and logical representations to determine the relation of textual entailment between the logical representations of the question and the text passage, | RB | Open-domain | Analyze questions, extract features of the questions, identify relevant passages based on features of the questions. | AQA-WebCorp | Accuracy = 74% | The study does not retrieve passages. |
| *Antoun, Baly & Hajj (2020a)* | AraBERT → It described the process of pretraining the BERT transformer model specifically for the Arabic language. | DL → AraBERT | Closed-domain | Two main phases: pre-train and fine-tune the model. | -TyDiQA<br>-ARCD | EM-TyDiQA = 71%<br>F1-TyDiQA = 83%<br>EM-ARCD = 31%<br>F1-ARCD = 65% | Improvements to RC are required in the pre-training phase. |
| *Antoun, Baly & Hajj (2020c)* | AraELECTRA → This model introduced different approach that uses the replaced token detection (RTD) objective instead of the traditional masked language modeling (MLM) objective used in AraBERT. | DL → Ara-ELECTRA | Closed-domain | Two main phases: pre-train and fine-tune the model. | -TyDiQA<br>-ARCD | EM-TyDiQA = 74%<br>F1-TyDiQA = 86%<br>EM-ARCD = 37%<br>F1-EM-ARCD = 71% | Improvements to RC are required in the pre-training phase. |
| *Antoun, Baly & Hajj (2020b)* | AraGPT2 → The paper presented the first advanced Arabic language generation model. The model is trained on a large corpus of internet text and news articles. | DL → AraGPT2 | Closed-domain | Two main phases: pre-train and fine-tune the model. | -TyDiQA<br>-ARCD | EM-TyDiQA = 3%<br>F1-TyDiQA = 14%<br>EM-ARCD = 4%<br>F1-EM-ARCD = 13% | The study is not suitable for QA systems as it generates texts rather than extracting answers. |
| *Zeid, Belal & El-Sonbaty (2020)* | N/A → The novelty of this paper is the development of an Arabic QA system using graph ontology and multiple semantic techniques. | RB with graph ontology | Open-domain | Three main steps: Question processing → remove stop words, use Khojah stemmer, and AWN. Document processing → use graph ontology, and Answer extraction → use Google search API | -Saudi Arabia Ministry of health<br>-TREC and CLEF<br>-Fatwaa Corpus (Islamic Religion datasets) | Precision = 76% Recall = 95%<br>Accuracy = 84%<br>C@1: 84.6% | Several processing steps are required for each question, document, and passage, which may delay answer retrieval. |
| *Alamir et al. (2021)* | N/A → This research paper introduced a QA system designed specifically for a Saudi Arabia labor law dataset. The key innovation of this study lies in addressing the challenge of constructing an efficient question-answering system tailored to the Arabic language. | RB | Open-domain | Three main stages: Data preparation, data processing, and answer extraction. TF-IDF and cosine similarity are used to retrieve answers. | Labor law dataset from Ministry of Human Resources and Social Development in the KSA | F1=0.06% | The system handles only those words that have the similarity in both question and document. |
| *Maraoui, Haddar & Romary (2021)* | N/A → The paper focused on a specific domain, *i.e.*, Islamic sciences, and uses a normalized database specified to retrieve accurate answers for factoid questions related to Hadith, narrator, and Tafsir text. | RB | Closed-domain | Three main phases: question analysis, information retrieval, and answer processing. | -Hadith<br>-Tafsir | Accuracy = 92% | There are several processing steps for each question, document, and passage which may cause delays in retrieving answers. |
| *Abdul-Mageed, Elmadany & Nagoudi (2021)* | ARBERT/ MARBERT → this study introduced the two models which are designed to serve a collection of diverse Arabic varieties. | DL → ARBERT and MARBERT | Closed-domain | Same method as that of BERT transformer model and AraBERT model. | -ARCD<br>-MLQA<br>-XQuAD<br>-TyDiQA | - ARBERT:<br>EM-TyDiQA =46%<br>F1-TyDiQA = 66%<br>EM-ARCD = 27%<br>F1-EM-ARCD = 60%<br>EM- XQuAD = 49%<br>F1- XQuAD= 67%<br>EM- MLQA= 34%<br>F1- MLQA= 53%<br>- MARBERT:<br>EM-TyDiQA = 38%<br>F1-TyDiQA = 57%<br>EM-ARCD = 23%<br>F1-EM-ARCD 55= %<br>EM- XQuAD= 41%<br>F1- XQuAD= 58%<br>EM- MLQA= 28%<br>F1- MLQA=45% | Process of model building was not different from existing models. Only the texts on which they were trained were different. |

Alrayzah et al. (2023), *PeerJ Comput. Sci.*, DOI 10.7717/peerj-cs.1633

**Table 4** (*continued*)

| Ref. | System or model/novelty | Approach | Domain | Methodology | Dataset | Evaluation result | Limitations/Future work |
|------|------------------------|----------|--------|-------------|---------|-------------------|------------------------|
| *Alrowili & Shanker (2021)* | ArabicTransformer → It combined both Funnel transformer with AraELECTRA as one model. | DL → ArabicTransformer: B6-6-6 and B4-4-4 | Closed-domain | Combine both funnel transformer and ELECTRA objective. | TyDiQA ARCD | For B6-6-6 TyDiQA F1: 87.21 EM: 75.35 ARCD F1: 72.70 EM: 36.89 For B4-4-4 TyDiQA F1: 85.89 EM: 74.70 ARCD F1: 67.70 | Only two QA datasets were tested. |
| *Nagoudi, Elmadany & Abdul-Mageed (2022)* | AraT5 → The paper proposed a novel unified benchmark for Arabic natural language generation (ARGEN) composed of seven NLP tasks. | DL → AraT5 | Open-domain → Text generation based on understating the given language (question generation) then answer questions. | Two phases: -fill-in-the-blank cloze to predict the tokens in the text. -Fine-tunning text generation. | - ARGEN$_{QG}$ | Bilingual Evaluation Understudy (Bleu) score= 16% | Reasons for low accuracy were not discussed. The model was not suitable for QA, especially in the closed domain. |
| *Alsubhi, Jamal & Al-hothali (2021)* | N/A → the introduced a system for open-domain based on DL algorithm to retrieve documents. | DL → DPR as retriever for IR and AraELCTRA as reader. | Open-domain | Three main phases: question analysis, information retrieval, and answer processing. | -ARCD -SQuAD -TyDiQA | -EM-TyDiQA = 74% -F1-TyDiQA = 86% -EM-ARCD = 37% -F1-EM-ARCD = 68% | QA dataset was not enough for training the system as it used low resource dataset (Arabic dataset). |
| *Hamza et al. (2022)* | N/A → This study analyze the behavior of ELMo representation by building numerous neural network architectures trained to classify questions and compares it to a context-free representation. | DL → using embeddings of many transformers model (BRT, ELMo) and AraBERT | Open-domain | Vanilla classifier used with feature-level fusion of three of word representations (BRT, ELMo and AraBERT) | Dataset for classifying the questions: -Moroccan school books. -CLEF -TREC | The result of question classifier: -F1 of AraBEERT with W2V= 93% -F1 of AraBERT with ELMo= 90% - AraBERT with Vanilla= 93% | The study does not apply to QA datasets. Authors are suggested to apply the proposed classification method on QA system for open domain questions. |
| *Premasiri et al. (2022)* | N/A → Developed a system to improve Arabic QA in Qur'an domain. | DL (transfer learning/ensemble learning) → AraELECTRA, camelbert, mbert, and AraBERT in different versions | Fine-tuning: Closed-domain | Ensemble learning is used to predicts one result from several Arabic transformer models. | - ARCD - QRCD (Qur'an Reading Comprehension Dataset) - SQuAD | partial Reciprocal Rank (pRR) = 49% | The paper presents overall scores of the transfer learning without results of QA datasets. |

Alrayzah et al. (2023), *PeerJ Comput. Sci.*, DOI 10.7717/peerj-cs.1633

**Table 4** (*continued*)

| Ref. | System or model/novelty | Approach | Domain | Methodology | Dataset | Evaluation result | Limitations/Future work |
|---|---|---|---|---|---|---|---|
| *ElKomy & Sarhan (2022)* | N/A →The paper proposed solution uses an ensemble learning model based on Arabic variants of BERT models to improve Qur'an QA. | DL →BERT, AraBERT, ARBERT, and MARBERT | Fine-tuning: Closed-domain | Ensemble method with voting approach is used. It collects all prediction results, and then merges them. Post-processing operations. | - QRCD | -Ensemble: F1= 59% EM= 39% pRR= 63% -Ensemble_POST: F1= 59% EM= 38% pRR= 65% | The stacking ensemble approach is recommended for use as a substitute to voting ensemble. |
| *Alnajjar & Hämäläinen (2022)* | N/A → The study proposed a method to predict answers to questions based on a passage of Qur'an. | DL → reimplement multilingual BERT | Fine-tuning: Closed-domain | Build a new model for Quaran QA by reimplementing multilingual BERT. The new model has the same setting of BER but with adding Islamic corpus. | - Cross-lingual Question Answering Dataset (XQuAD) - QRCD - SQuAD - Multilingual Question Answering (MLQA) Summing all dataset to obtain one large dataset. They score: | For model: partial Reciprocal Rank (pRR) = 34% For dataset: - pRR= 70% -F1= 35% -EM= 68% | The paper does not contain any information about the size of new corpus. |
| *Aftab & Malik (2022)* | N/A → This study improved the BERT output by applying regularization techniques like weight-decay and data augmentation. | DL → BERT | Fine-tuning: Closed-domain | -Reimplement the BERT model and fine-tuning it. - Use regularization techniques: data augmentation and wight-decay. | QRCD | - pRR= 58% -F1= 56% -EM= 31% | The results have to be enhanced by increasing training on QA dataset. |
| *Mostafa & Mohamed (2022)* | N/A → This proposed a method for efficient QA for the Qur'an in the Arabic language using DL approach. | DL → Ara-ELECTRA | Fine-tuning: Closed-domain | -Use AraELECTRA. -Try different loss functions to find the best function for increasing the performance of the model. -Train the model on several Arabic QA datasets. | QRCD | - pRR= 66% | Authors are suggested to increase the dataset size to improve the robustness of the model. |
| *Touahri et al. (2022)* | N/A → This study developed a Transformer-based QA system using the mT5 Language Model for finding answers to questions from the Qur'an. | DL → Sequence-to-sequence model (mT5) | Fine-tuning: Closed-domain | -Apply mT5 sequence—to-sequence model. | QRCD | The best result was when using mT5-XL. Result on Dev set: - pRR= 98% - F1= 97% -EM= 98% Result on test set: - pRR= 43% - F1= 20% -EM= 40% | Reduced performance results were observed. |

Alrayzah et al. (2023), *PeerJ Comput. Sci.*, DOI 10.7717/peerj-cs.1633

**Table 4** (*continued*)

| Ref. | System or model/novelty | Approach | Domain | Methodology | Dataset | Evaluation result | Limitations/Future work |
|---|---|---|---|---|---|---|---|
| *Alsaleh, Althabiti & Al-shammari (2022)* | N/A → This paper used three Arabic pre-trained language models (AraBERT, CAMeL-BERT, ArabicBERT) for Qur'an QA. | DL → AraBERT, CAMeL-BERT, Ara-bicBERT | Fine-tuning: Closed-domain | Fine-tune the three models on QRCD | QRCD | - pRR= 45%<br>- F1= 42%<br>-EM= 16% | Authors are suggested to use ensemble learning to improve the performance of models. |
| *Singh (2022)* | N/A → The novelty of this paper lies in the proposed techniques used to solve the task, namely conditional text-to-text generation, embedding clustering, and transformers-based QA. | DL → AraBERT, BERT, mBERT, and mT5 | Fine-tuning: Closed-domain | Use three techniques that known as semantic embeddings and clustering, Seq2Seq based text span extraction, and fine-tuning BERT model. | QRCD | For first solution:<br>- pRR= 23%<br>- F1= 11%<br>-EM= 2%<br>For second solution:<br>- pRR= -%<br>- F1= -%<br>-EM= 30%<br>For third solution:<br>- pRR= 25%<br>- F1= 13%<br>-EM= 8% | The first phase of transformer models, related to reading comprehension has to be improved |
| *Keleg & Magdy (2022)* | N/A → The study proposed creating better faithful splits from the original dataset and fine-tuning a model on classical Arabic text, which yielded the best performance on the new evaluation split. | DL → Arabic BERT | Fine-tuning: Closed-domain | -Categorize dataset based on question types. -Build faithful splits to generate new training and development dataset. -Detect data leakage between two splits of the dataset. | QRCD | - pRR = 40% | The size of the dataset was not enough for training |

**Notes.**

RB, rule-based; ML, machine learning; DL, deep learning; Ref., reference.

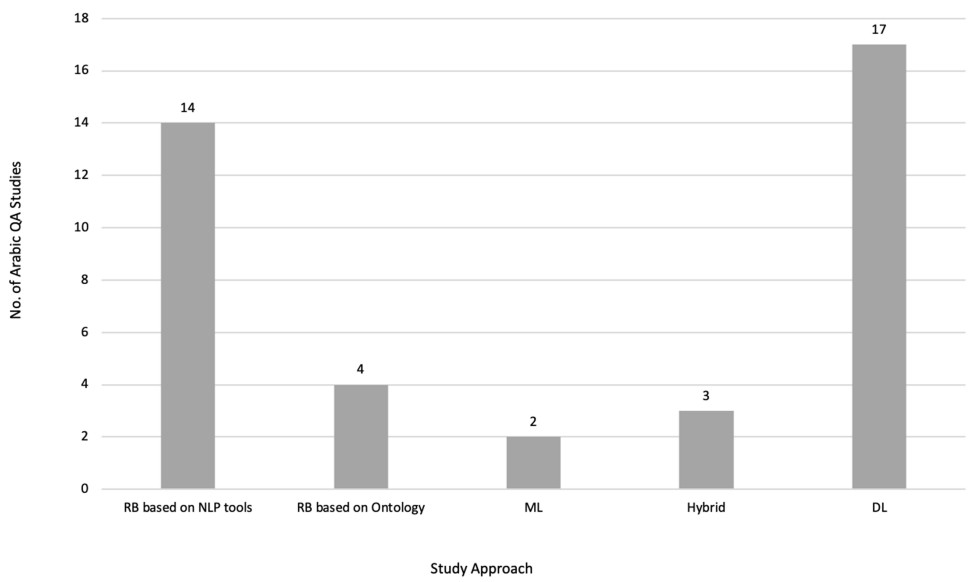

**Figure 6** **Distribution of Arabic QA studies based on used approach.**

Seventeen published articles suggested the enhancement of Arabic QA tasks using DL, followed by RB approach, which was suggested from 2013 until 2018, as shown in Fig. 6. Figure 7 illustrates the distribution of Arabic QA studies based on the aimed domains. Approximately 29 studies suggested for open-domain, which was approximately 58% of all the studies, while 10 studies suggested Arabic closed-domain, which was approximately 22% of all the studies. The remaining studies were based on excising systems suggested for closed-domain by fine-tuning technique. The models were trained on the QA dataset.

A majority of the QA systems have question analysis, document retrieval, and answer extraction as three main phases in common. Each component should follow at least one of the steps illustrated in Fig. 8.

### RQ2: Which are the Arabic QA datasets used to evaluate the Arabic QA systems and models proposed by the researchers, and which datasets are available to the public?

Several Arabic QA datasets are available for research and development. As listed in Table 4, numerous available Arabic QA datasets are different based on the covered domain. Some of the examples are as follows.

The Arabic SQuAD 2.0 serves as an Arabic adaptation of the renowned Stanford question answer dataset (SQuAD), which is extensively employed for QA model assessment and training (_Mozannar et al., 2019_). The dataset contains more than 100,000 questions and answers on a variety of topics.

Approximately 1,395 questions on Wikipedia articles are available in the Arabic reading comprehension dataset (ARCD) (_Mozannar et al., 2019_).

The Arabic question-answer dataset (AQAD), a vast collection of question-answer pairs designed for Arabic reading comprehension, is a recent addition to the dataset landscape

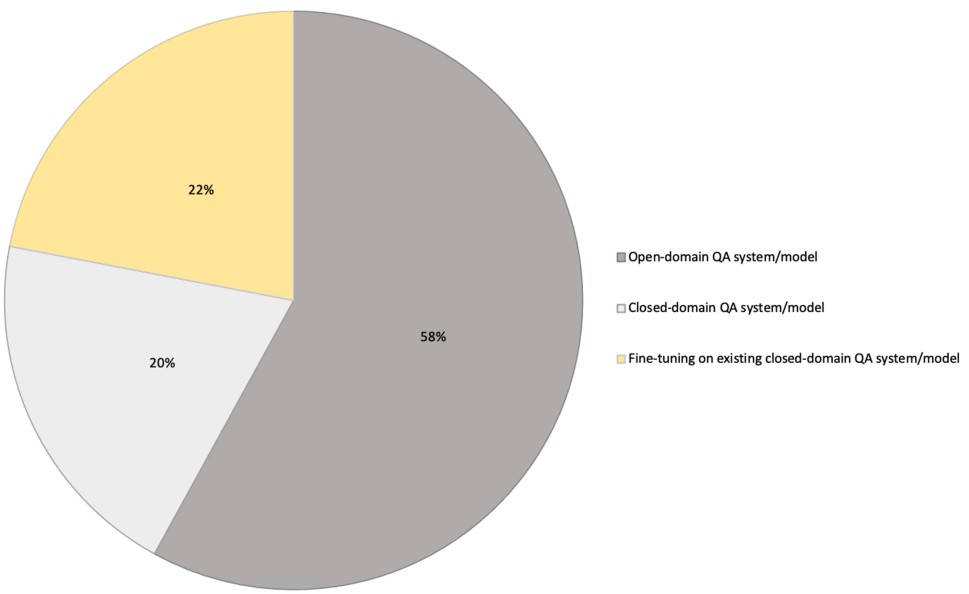

**Figure 7** Distribution of Arabic QA studies over different domains.

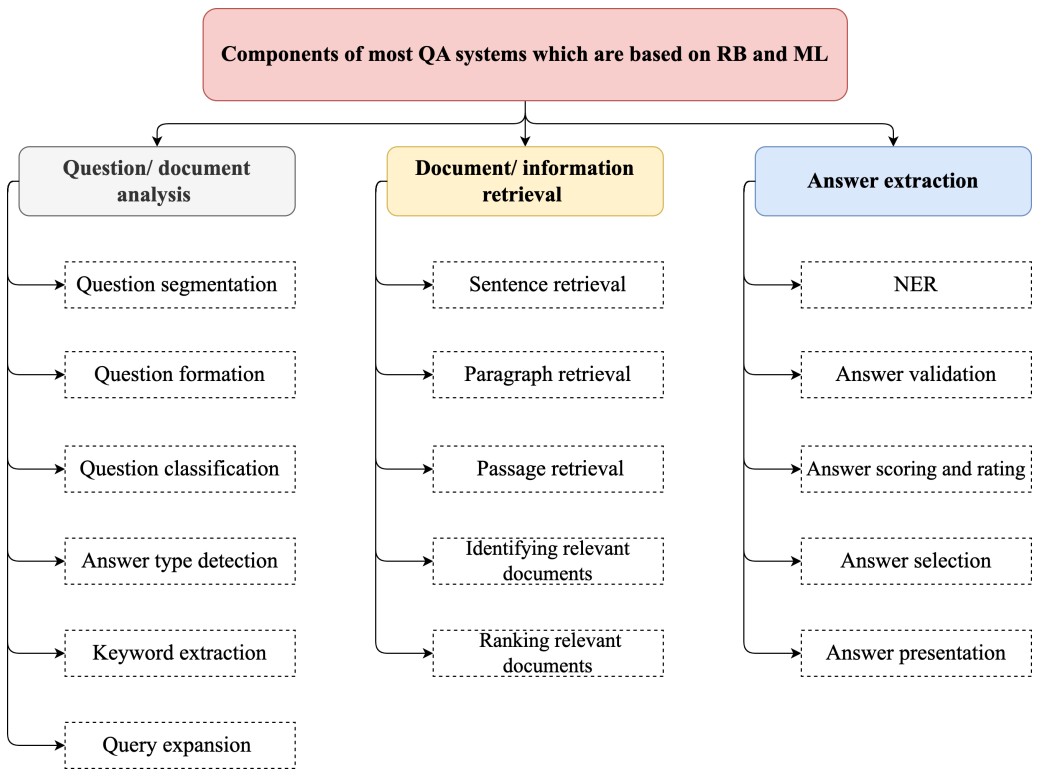

**Figure 8** General components and steps of most common Arabic QA studies.

(*Atef et al., 2020*). With over 17,000 meticulously crafted questions and answers, this dataset is of exceptional quality.

The dataset for Arabic "why" QA system (DAWQAS) is a dataset specifically created for Arabic "why" questions, consisting of 3,205 pairs of "why" questions-answers across eight domains. The questions were sourced from six public Arabic websites, resulting in a total of 3,259 pages, with multiple ranked answers. After removing redundant questions and those with multimedia responses, the dataset was refined to 3,205 "why" QA pairs, containing 44,843 answer sentences (*Almarwani & Diab, 2018*).

The Arabic QA web corpus (AQA-WebCorp) was created by querying the Google search engine. The dataset comprises 250 question and text pairs, with some questions remaining unanswered or answered inaccurately. The data were gathered from online sources (*Bakari, Bellot & Neji, 2016a*).

The Arabic Qura'n question (AQQAC) is a vast compilation of more than 2,200 entries that pertain to inquiries and responses about the Holy Qur'an. Each entry is equipped with pertinent metadata, such as the question ID, chapter number, verse number, question topic, question type, and Al-Qur'an ontological concepts. Moreover, the origin of each inquiry is also included.

Multi-lingual QA (MLQA) is a large-scale multi-lingual extractive QA dataset with samples paralleled between four different languages on an average. The dataset contains a total of 12,000 and more than 5,000 QA instances in SQuAD format for English and extractive QA instances for other six languages, respectively (*Lewis et al., 2020*).

QRCD comprises 1,093 pairs of questions and passages, along with their corresponding extracted answers, resulting in a total of 1,337 triplets of question–passage–answer (*Malhas & Elsayed, 2022*).

OR-QA retrieve is a task that involves finding the English document that best answers a question written in another language (example, Japanese). The dataset for this task is named as XOR-QA retrieve. It is a collection of questions and answers in seven different languages. The questions were written by people seeking information, while the answers were collected from a multi-lingual collection of documents (*Asai et al., 2021*). The dataset is designed to support cross-lingual answer retrieval, which means finding answers in one language to questions written in another language.

Other Arabic QA datasets such as ANERcorp, cross-lingual question answering dataset (XQuAD) (*Artetxe, Ruder & Yogatama, 2020*), and TyDiQA (*Clark et al., 2020a*) are also available. The first complex Arabic QA dataset was published by *Sidhoum et al. (2023)*. The dataset is an Arabic complex question answering dataset (ACQAD) that includes more than 118,000 questions, encompassing both comparison and multi-hop types. More than one passage or document has to be read for extracting the answers.

The aforementioned datasets are the most common Arabic QA datasets available for training and evaluating QA models. They can also be used as a benchmark to compare the performance of different models.

Data provided in Table 5 presents a summary of the most commonly used Arabic QA dataset for evaluating the proposed solutions. The summary is based on the name of the dataset, year of its creation, format, size, type of question in the corresponding dataset,

Alrayzah et al. (2023), *PeerJ Comput. Sci.*, DOI 10.7717/peerj-cs.1633

**Table 5  Most common Arabic QA datasets.**

| Dataset | Format | Size | Type | Translated? | Paper | Ref. | Link of access | Covered domain |
|---|---|---|---|---|---|---|---|---|
| ArabiQA 2007 | Q, A, D | 200 Q on 11,000 D | Factoid | X | Implementation of the ArabiQA Question Answering System's Components | Benajiba, Rosso & Lyhyaoui (2007) | N/A | Open-Wikipedia articles |
| DefArabicQA 2010 | Q, A, D | 50 Q | Defined | X | DefArabicQA: Arabic Definition Question Answering System | Trigui, Belguith & Rosso (2010) | N/A | Open-Wikipedia articles and Google search engine |
| TREC 2010 | Q, A | 1,500 | Factoid | Human Translator | An evaluated semantic query expansion and structure-based approach for enhancing Arabic question/answering | (Abouenour et al., 2010) | http://trec.nist.gov/data/qa.html | Open-General |
| CLEF 2010 | Q, A | 764 | Factoid | Human Translator | An evaluated semantic query expansion and structure-based approach for enhancing Arabic question/answering | (Abouenour et al., 2010) | http://www.clef-campaign.org | Open-General |
| QAM4MRE 2011 | Q, multiples A, D | 160 Q, D 800 Multiples Q, with 5 A for each Q | Factoid Multiple-choice | Expert Translator | Overview of QA4MRE at CLEF 2012: Question Answering for Machine Reading Evaluation | Peñas et al. (2012) | https://www.kaggle.com/datasets/thedevastator/harness-the-challenges-of-qa4mre-in-your-researc | Closed-Biomedical Texts about Alzheimer's disease, AIDS, Climate Change, Music and Society |
| QArabPro 2011 | Q, A, D | 335 Q, 75 D | Factoid Why How | X | QArabPro: A Rule Based Question Answering System for Reading Comprehension Tests in Arabic | Akour et al. (2011) | N/A | Open-Wikipedia articles |
| AQA-WebCorp 2016 | Q, D | 250 Q, D | Factoid | X Created by querying the search engine Google | AQA-WebCorp: Web-based Factual Questions for Arabic | Bakari, Bellot & Neji (2016a) | N/A | Open-sport, history & Islam, discoveries & culture, world news, health & medicine |
| DAWQAS 2018 | WhyQ, A, S | 3,205 whyQ, A On 44,843 S | Why | X Created by Google Search API | DAWQAS: A Dataset for Arabic Why Question Answering System | Almarwani & Diab (2018) | https://github.com/masun/DAWQAS | Closed- Sport, Politic, Arts & Celebrities, Technology & Science, Religion, Nature & Animals, Society & Women, and Health & Nutrition |
| AQQAC 2018 | Q, A | 2,224 Q, A | Factoid | X Extracted from the Altabari Tafseer | No paper → Alqahtani, Mohammad and Atwell, Eric (2018) Annotated Corpus of Arabic Al-Qur'an Question and Answer. University of Leeds | N/A | https://archive.researchdata.leeds.ac.uk/464/ | Closed- Islamic (Qura'n) |
| ARCD 2019 | Q, A, P | 1,395 | Factoid | X By crowd workers | Neural Arabic Question Answering Hussein | Mozannar et al. (2019) | https://huggingface.co/datasets/arcd | Closed- Arabic Wikipedia articles |
| Arabic-SQuAD 2019 | Q, A, P | 48,344 Q, A on 10,364 P | Factoid | Machine Translation | Neural Arabic Question Answering Hussein | Mozannar et al. (2019) | https://metatext.io/datasets/arabic-reading-comprehension-dataset-(arcd) | Closed- Wikipedia articles |
| AQAD 2020 | Q, A, P | 17,911 Q, A on 3,381 P | Factoid | X By data collector | AQAD: 17,000+ Arabic Questions for Machine Comprehension of Text | Atef et al. (2020) | https://github.com/adelmeleka/AQAD/tree/master/AQQAD%201.0 | Closed- Arabic Wikipedia articles |

Alrayzah et al. (2023), *PeerJ Comput. Sci.*, DOI 10.7717/peerj-cs.1633

**Table 5** (*continued*)

| Dataset | Format | Size | Type | Translated? | Paper | Ref. | Link of access | Covered domain |
|---|---|---|---|---|---|---|---|---|
| TyDiQA 2020 | Q, A | 15,429 Q, A, P | Factoid | X The questions performing by a Google search on the question text | TYDIQA: A Benchmark for Information-Seeking Question Answering in Typologically Diverse Languages | *Clark et al. (2020a)* | https://github.com/WissamAntoun/Arabic_QA_Datasets | Closed- Wikipedia articles |
| MLQA 2020 | Q, A, P | 5,852 Q, A, P | Factoid | ✓ Machine Translation | MLQA: Evaluating Cross-lingual Extractive Question Answering | *Lewis et al. (2020)* | https://github.com/facebookresearch/MLQA | Closed- Wikipedia articles |
| Ar-XQuAD 2020 | Q, A, P | 98,479 Q, A, P | Factoid | ✓ Professional Translator | On the Cross-lingual Transferability of Monolingual Representations | *Artetxe, Ruder & Yogatama (2020)* | https://www.tensorflow.org/datasets/catalog/xquad | Closed- Wikipedia articles |
| AyaTEC 2020 | Q, A, P | 207 Q, 1,762 A, 11 topics | Factoid | X | AyaTEC: Building a Reusable Verse-Based Test Collection for Arabic Question Answering on the Holy Qur'an | *Malhas & Elsayed (2020)* | http://qufaculty.qu.edu.qa/telsayed/datasets | Closed- Islamic (Qura'n) |
| Hadith and Tafsir 2021 | Q | 100 Q | Factoid | X Created by native Arabic speakers and online forums. | Arabic factoid Question-Answering system for Islamic sciences using normalized corpora | *Maraoui, Haddar & Romary (2021)* | N/A | Closed- Islamic sciences (Tafsir and Hadith) |
| Labor law dataset 2021 | Q, D | 100 Q | Factoid | X Extracted from Ministry of Human Resources and Social Development in the KSA | Arabic Question-Answering System Using Search Engine Techniques Manal | *Alamir et al. (2021)* | N/A | Open- Ministry of Human Resources and Social Development in the KSA |
| XOR-QA 2021 | Q, A, D | 5,235 Q | Factoid | Machine Translation | XOR QA: Cross-lingual Open-Retrieval Question Answering | *Asai et al. (2021)* | https://github.com/AkariAsai/XORQA | Open- Wikipedia 2019-0201 dump |
| QRCD 2022 | Q, A, P | 1,337 Q, A, P | Factoid | X | Arabic machine reading comprehension on the Holy Qur'an using CL-AraBERT | *Malhas & Elsayed (2022)* | https://github.com/RanaMalhas/QRCD | Closed- Islamic on Holy Qur'an |
| ACQAD 2023 | Q, A, D | 118K Q, A | Multi-hop Comparing | X Collected by Wikipedia API and BeautifulSoup library | ACQAD: A Dataset for Arabic Complex Question Answering | *Sidhoum et al. (2023)* | Contact authors | Open-Wikipedia |

**Notes.**

Q, Question; A, Answer; P, Paragraph; D, Document; S, Sentence; N/A, not available.

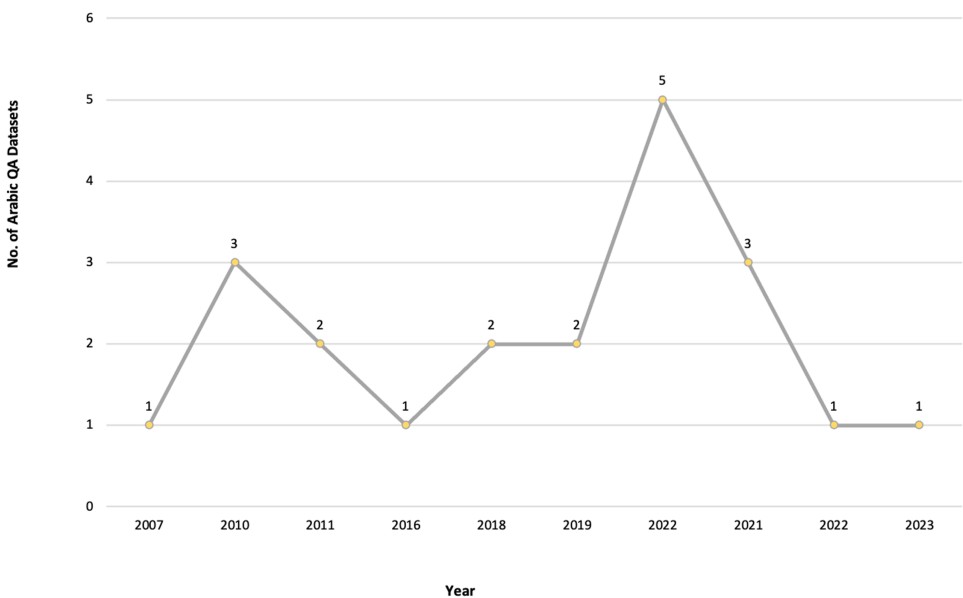

**Figure 9** Year-wise distribution of Arabic QA datasets.

whether original or translated, paper name with its reference, link of accessibility, and covered domain. As a result of Table 5, main advantage of those Arabic datasets is the free availability. Many Arabic QA datasets are available for free, making them accessible to researchers and developers without any cost. Also, some Arabic datasets have been created and curated by Arab researchers, leading to a better understanding of the linguistic and cultural nuances of the Arabic language.

Some Arabic QA datasets cover different domains such as general, biomedical, Islamic, sports, and histories. On the other hand, Arabic QA datasets suffer from some drawbacks. One of the main drawbacks is the relatively small size of many Arabic datasets. For training DL models, a substantial amount of data is required, often more than 100,000 samples. Additionally, many of Arabic QA dataset may contain ambiguous questions causing the machine translation leading to challenges in building reliable models.

Figure 9 shows the distributions of Arabic QA datasets over the years. A majority was observed in 2022 because the DL approach gained much attention. The distributions of Arabic QA datasets based on the domains and their originality and type of questions are shown in Figs. 10 and 11, respectively. The distribution approximately equally-covered both domains, but factoid-type questions was more popular than other questions, such as "why" questions.

## RQ3: Which evaluation criteria were utilized to assess the Arabic QA systems and models?

Assessing the effectiveness of QA systems is essential to ascertain their precision and competence in generating precise answers to questions in testing datasets. Several evaluation methods, such as precision, recall, mean reciprocal rank (MRR), pRR, exact match (EM),

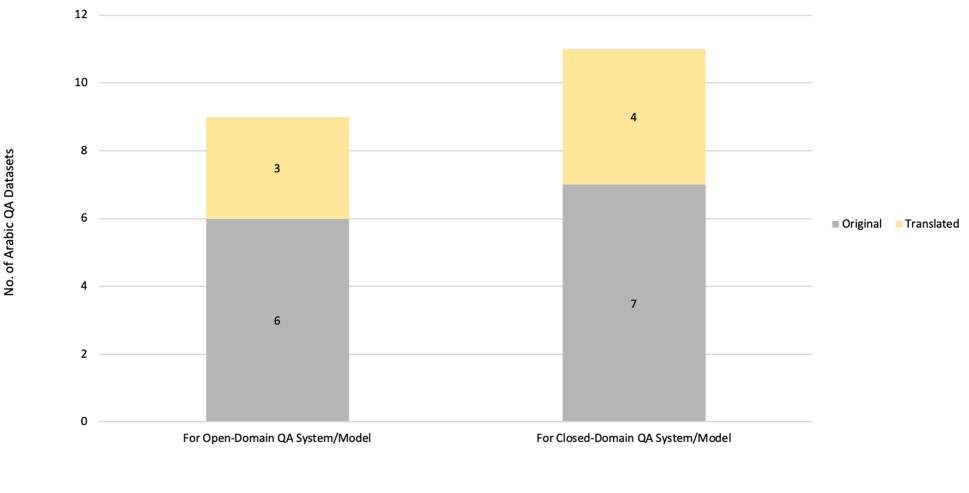

**Figure 10  Distribution of Arabic QA datasets based on the domains and their originality.**

accuracy, F1-score, and C@1, were utilized in related studies to appraise the performance of QA systems. A detailed discussion on these measures are as follows.

### Precision

Precision is a measure of number of query-relevant documents retrieved in a search. To calculate precision, the number of useful documents is divided by the total number of documents retrieved. For closed-domain, precision measures the percentage of number of correct answers answered over total number of all questions answered or number of correct answers/total number of questions answered (*Alwaneen et al., 2021*).

### Recall

Recall is a metric that quantifies the ability of a system to retrieve correct answers. To calculate recall, the number of correct answers retrieved by the system is divided by the total number of questions selected for retrieval or number of correct answers/number of selected questions to be answered (*Alwaneen et al., 2021*). The range of both precision and recall has to be between 0 and 1 as percentage (*Biltawi, Tedmori & Awajan, 2021*).

### Accuracy

Accuracy is a measure of the number of correctly answered questions, expressed as a percentage of the total number of questions (*Abdel-Nabi, Awajan & Ali, 2022*). It can be calculated as, number of correct answers/total number of questions. For open-domain QA systems, accuracy measures the effectiveness of a search engine (information retriever) in locating the desired documents. It is calculated as the fraction of correctly retrieved documents (those that are relevant to the query) divided by the total number of documents fetched by the search engine (*Biltawi, Tedmori & Awajan, 2021*).

### Mean reciprocal rank

MRR evaluates the quality of a QA system by taking the reciprocal of the rank of the first correct answer in the list of retrieved answers from the system (*Alwaneen et al., 2021*; *Biltawi, Tedmori & Awajan, 2021*). It is computed by first calculating the rank of the first result for each query, and then taking the mean of all the ranks.

### Partial reciprocal rank

The pRR metric is a modified version of the RR metric, specifically designed for assessing the effectiveness of IR systems. pRR measures the quality of the first retrieved answers (k), where k is a user-specified parameter (*Calijorne Soares & Parreiras, 2020*). pRR is calculated as the average reciprocal rank of the first k answers that are relevant to the given query.

### Exact match

EM is a measure of closeness of a predicted answer to the ground truth answer, where 1 indicates a perfect match (*Abdel-Nabi, Awajan & Ali, 2022*).

### F1-score/F1-measure

The F1-score, a widely used metric for evaluating models in QA systems, is derived from precision and recall. It is computed as the harmonic mean of precision and recall and can be expressed as, (2×precision ×recall) / (precision+recall) (*Alwaneen et al., 2021*; *Biltawi, Tedmori & Awajan, 2021*).

### Correct at 1 (C@1)

Correct at 1 or C@1 was introduced in 2011 to decrease the count of incorrect answers while upholding the number of accurate ones by choosing answers to certain questions rather than compelling them to give an incorrect answer (*Peñas & Rodrigo, 2011*).

## RQ4: What are the current research methods and status of Arabic QA models and systems?

On analyzing the selected primary studies and summarized data illustrated in Table 2, Figs. 6 and 7, and dataset distributions in Figs. 10 and 11, the DL approach, especially PLMs, was observed to be relevant in recent years from 2022. Hence, further details on PLMs and Arabic PLMs were analyzed and are presented in this section.

### Pre-trained language models (PLMs)

*Architecture of PLMs.* PLMs or large neural networks have been trained on extensive data and can be applied to various NLP tasks. They follow a two-step process, where they first pre-train a model on a vast text corpus, and then fine-tune the model on a downstream task using small labeled datasets (*Elazar et al., 2021*). PLMs are recognized as proficient language encoders, offering fundamental language comprehension abilities that can be utilized for diverse downstream tasks (*Elazar et al., 2021*). Recently, PLMs have gained significant traction in NLP, exhibiting exceptional performance on numerous tasks, and have emerged as a preferred choice (*Min et al., 2021*).

Pre-training is inspired by the learning behavior of human beings, who transfer and reuse old knowledge to understand new knowledge and handle new tasks. Similarly, PLMs

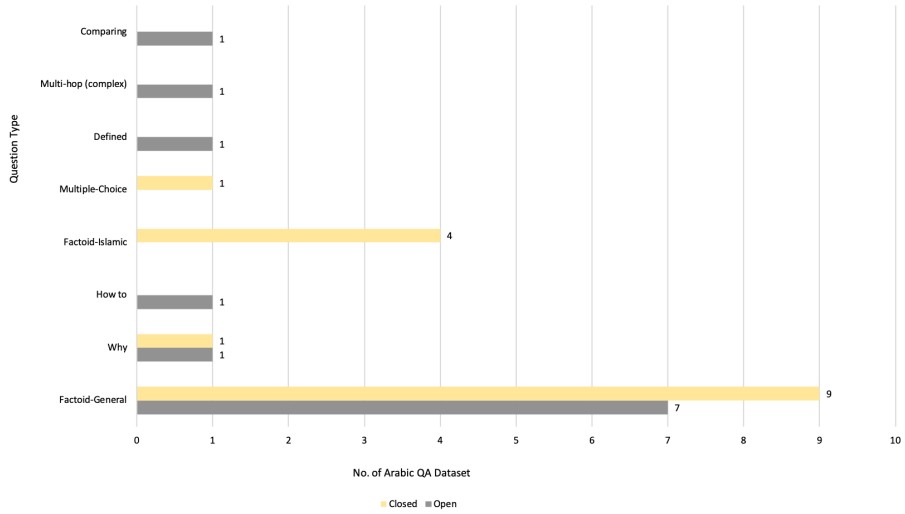

**Figure 11** Distribution of Arabic QA datasets based on the domain and type of questions.

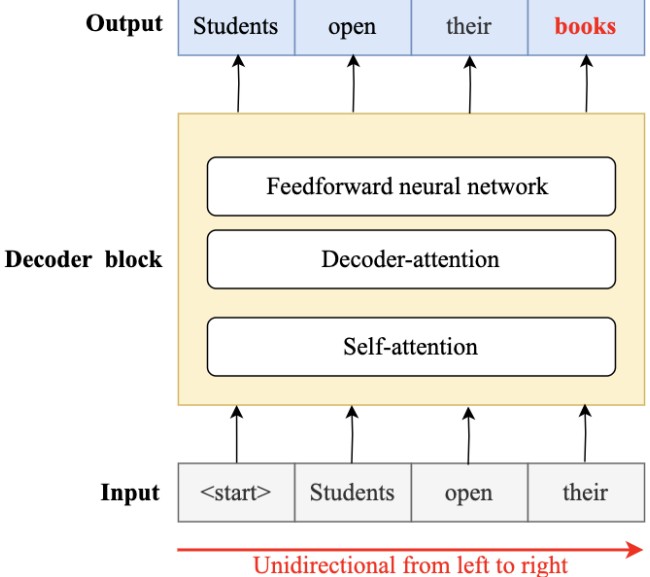

**Figure 12** Decoder (autoregressive/CLM) language models.

are capable of successfully completing new tasks using their old experiences and knowledge (*Li et al., 2021*). PLMs are also known as transformers because knowledge is transferred from the pre-training phase to the fine-tuning phase to perform several downstream tasks without any requirement of re-tarring transformers to understand languages.

A transformer is a type of neural network that uses attention and self-attention to create a stack of encoders and decoders. Based on existing studies (*Alyafeai, AlShaibani & Ahmad, 2020*; *Kalyan, Rajasekharan & Sangeetha, 2021*; *Li et al., 2021*; *Min et al., 2021*),

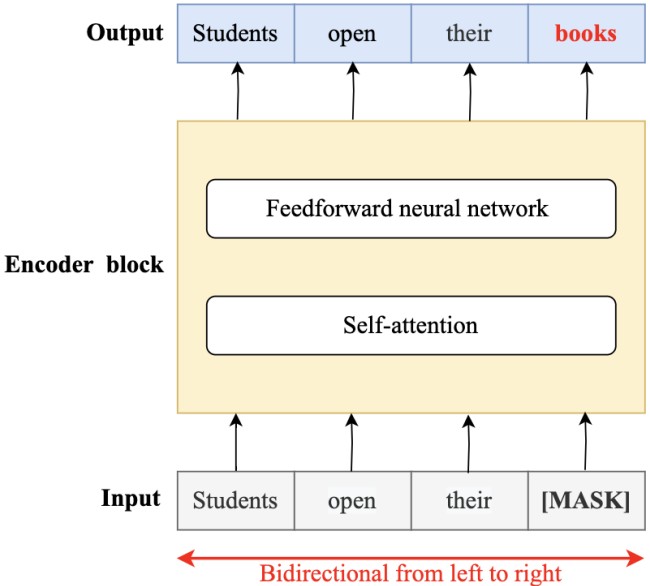

**Figure 13** Encoder (masked) language models.

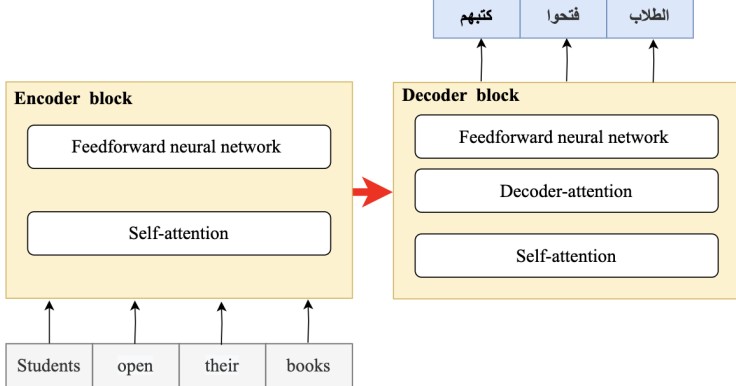

**Figure 14** Encoder–decoder (sequence-to-sequence) language models.

three main architectures of PLMs were identified based on the direction of processing sequence and pre-training tasks. These architectures are illustrated in Figs. 12, 13 and 14, and include decoder (unidirectional), encoder (bidirectional), and encoder–decoder (unidirectional–bidirectional) PLMs.

*Decoder-based models* Decoder-based transformers are unidirectional models that process sequence from left to right. They are also known as auto-regressive language model or casual language modeling (CLM) owing to the corrupt document rotation strategy followed. These models are trained to predict the next word (xi) in a sequence, given all of the previous words (x1, x2, ..., xi-1) in the sequence (*Min et al., 2021*). Decoder-based transformers

have blocks consisting of feedforward neural networks (FNN), which are the same RNN, but without return to back, self-attention, and decoder attention. The self-attention layer allows tokens to look at each other. After taking information from other tokens, the model takes a moment to think and process the information using FNN. The latter layer assists the model to pay attention to specific segments from the input sequence. Stacking multiple transformer-decoder layers is possible with masked self-attention in order to improve performance. This allows the model to consider all previous tokens in the sequence when making a prediction for the next token (*Wolf et al., 2020*). Auto-regressive models are invented to generate texts conditioned by the input text. However, these models lack independent modules for encoding the input sequence, which limits their practicality (*Li et al., 2021*).

As shown in Fig. 12, tokens that are close to the current context are only taken into account. The left context consists of the previous two items in the sequence (x1 and x2), while the right context consists of the next item in the sequence (x4). This flow allows the prediction of item (x3) from the left and right contexts only (*Alyafeai, AlShaibani & Ahmad, 2020*). Thus, these types of transformers are called auto-regressive language models.

*Encoder-based models* Encoder-based PLMs work in two directions (bidirectional) because they process sequence from left to right and right to left. Every token (element) in a given context can be attended by any other token in that same context (*Alyafeai, AlShaibani & Ahmad, 2020*). When (x3) appears in the context of (x1, x2, x4), it sees the whole context. Encoder-based models are known as masked language models. They randomly mask some tokens in the input sequence in order to improve the learning ability of the model from the sequence (*Yulita et al., 2023*). Predicting the masked tokens is the main objective of these models. The tokens that are masked are labeled [MASK], as shown in Fig. 13. For example, the following representation (x1, x2, [MASK], x4) would predict the masked token (x3) (*Alyafeai, AlShaibani & Ahmad, 2020*). This approach encourages the model to consider information from both directions when making predictions. Hence, the approach is also known as masked language modeling (MLM) (*Nassiri & Akhloufi, 2022*).

Models that use masked self-attention are capable of learning representations that are more complex and meaningful than those that do not use masked self-attention because masked self-attention allows a model to attend to all other tokens in a sequence, in both directions, when learning a representation for a particular token. This masking approach allows parallel computation, which is often more efficient at inference time (*Min et al., 2021*).

In addition to MLM, another step known as next sentence prediction (NSP) is used in some of the masked models. The objective of the NSP is to predict whether a given sequence A is followed by a given sequence B. The training of such a task consists of two sequences at each iteration. Sentence A is followed by sentence B 50% of the time (*Nassiri & Akhloufi, 2022*).

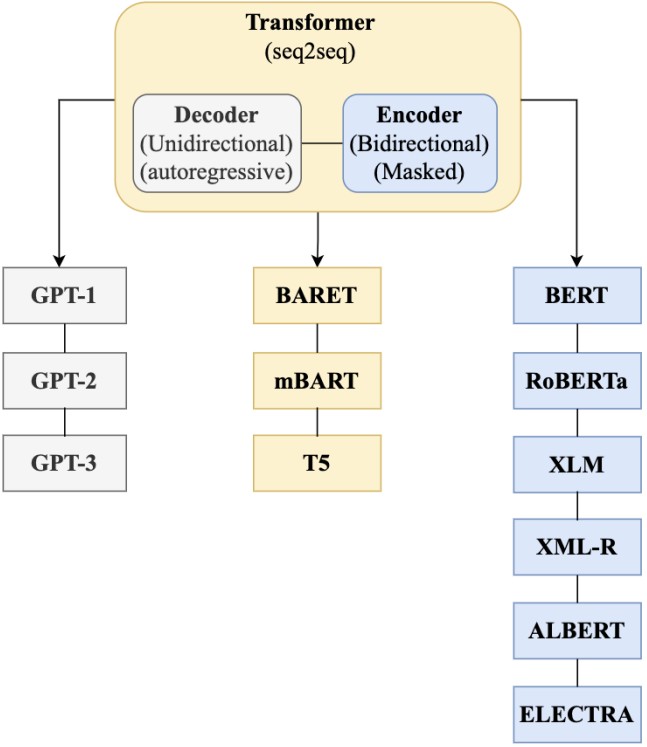

**Figure 15  Examples of transformer models.**

_Encoder–decoder based models_  An encoder–decoder model is a more flexible approach to text generation because it is capable of learning the sequence generation of tokens (y1, …, yn), given an input sequence (x1, …, xm). These type of models are also known as sequence-to-sequence (Seq2Seg) language models and are used to generate texts based on input text (sequence/context) (_Alyafeai, AlShaibani & Ahmad, 2020_). Encoder–decoder models consist of both encoder and decoder blocks that work as unidirectional and bidirectional transformers, as shown in Fig. 14. A Seq2Seq model, which utilizes an encoder to encode the input sequence and a decoder with an attention mechanism to generate the corresponding output sequence, is particularly effective for tasks such as machine translation, text summarization, and style transfer because the model is specifically designed to learn the mapping between two sequences (_Min et al., 2021_).

Examples of the three main types of PLMs, summarized in Fig. 15, were collected from several existing studies (_Kalyan, Rajasekharan & Sangeetha, 2021_; _Li et al., 2021_; _Min et al., 2021_; _Casola, Lauriola & Lavelli, 2022_). All examples and a majority of the models were designed to process English language. Only a few models were developed for processing Arabic, and the same have been discussed in the following sections.

_Pre-training tasks of PLMs._  All transformers, also known as PLMs, have to be pre-trained to large corpora in different languages based on the goals of the models. Large corpora are

**Table 6   Summary of pre-training tasks in PLMs.**

| Pre-training Task | Description |
| --- | --- |
| MLM | Initially, the model masks certain tokens within the input sequences and subsequently learns to predict the masked tokens by relying on the remaining ones. Examples: **BERT** |
| NSP | Training corpus is used to teach the model to recognize continuous and discontinuous segments in input sequences. Example: **BERT** |
| SOP | A pair of consecutive sequences from a document are used to train the model, along with a pair of identical sequences from the same document with their order reversed. This allows the model to learn the sequence order and identify when two sequences are swapped. The NSP is trained to predict the accurate arrangement of sequences within a text based on an input that is arranged in a random order. Example: **ALBERT** |
| CLM | The model predicts the last token in an input sequence by following the process of autoregressive models based on generating text. Example: **GPT-1/2/3** |
| RTD | The context in which a token appears is used by the model to determine its replacement. Example: **ELECTRA** |
| E-MLM | Models are allowed to predict masked tokens in dynamic style rather than in static style. Example: **RoBERTa.** |
| SPAN-MLM | MLM developed by masked contiguous of token rather than a single token. Example: **SpanBERT.** |
| PLM | From among all possible permutations, a target is picked at random. The targets are selected from the permuted sequence, and the model is trained to make predictions about those tokens based on the remaining tokens and the positions of the targets in the sequence. Example: **XLNet.** |
| Seq2Seq MLM | An auto-regressive decoder takes a masked sequence as input and produces masked tokens one at a time. Example: **T5.** |

collected from several different resources such as articles, news, books, social media, web crawl, and Wikipedia. A variety of pre-training tasks, such as MLM, NSP, CLM, replaced token detection (RTD), Seq2Seq MLM, exist based on encoder–decoder architecture (*Kalyan, Rajasekharan & Sangeetha, 2021*), enhanced masked language modeling (E-MLM) or dynamic masking instead of static masking (MLM), permuted language modeling (PLM), and sentence order prediction (SOP) (*Qiu et al., 2020*). Each model is pre-trained to specific tasks based on aimed downstream tasks of the model. MLM with SPAN (contiguous words) instead of single token, leads to a more accurate result of the-state-of-art PLMs (*Benlahbib, Alami & Alami, 2021*; *Glass et al., 2020*; *Joshi et al., 2020*; *Ram et al., 2021*). Table 6 presents a brief discussion on each pre-training tasks in transformers. All PLMs or transformers have the same two phases of training models: pre-training and fine-tuning phase. The differences are in terms of size and type of corpora, number of hyperparameters (hidden layers, epochs, batches, learning rate, attention heads, and activation function), and type of architecture.

*Arabic per-trained transformer models.* The introduction of PLMs proved to be a revolutionary development in the field of NLP, and the use of BERT resulted in state-of-the-art performance on several NLP tasks, such as QA (*Antoun, Baly & Hajj, 2020a*). BERT and similar models have the advantage of being pre-trained on large amounts of data, which the model can use to learn language representations (*Khurana et al., 2022*).

The integration of PLMs has presented a significant breakthrough in the field of NLP research, resulting in exceptional performance of various NLP tasks by transformer models such as BERT, RoBERTa, GPT, XML, and ELECTRA. Initially, a majority of these models were designed only for the English language, but eventually, they were adapted for other languages as well. In recent years, a number of transformer models have been designed for the Arabic language (*Antoun, Baly & Hajj, 2020a*). However, only a few direct comparisons between the different proposed models are available. In this section, the architecture and tasks of Arabic models are discussed.

AraBERT (*Antoun, Baly & Hajj, 2020c*) was the first transformer model developed for the Arabic language, and its introduction has assisted in improving the performance of several Arabic NLP tasks. Recently, numerous Arabic transformer models have been released, which include BERT based models such as versions of AraBERT, ARBERT/MARBERT (*Abdul-Mageed, Elmadany & Nagoudi, 2021*), and Arabic BERT (QARiB) (*Abdelali et al., 2021*). Other models with Arabic variants, such as AraGPT2 (*Antoun, Baly & Hajj, 2020c*), AraELECTRA (*Abdelali et al., 2021*), and Arabic ALBERT (*Alsubhi, Jamal & Alhothali, 2021*; *Antoun, Baly & Hajj, 2020c*), have also been released. The architectures, sizes, and training data of the aforementioned models vary. Although a majority of these models are trained on MSA data, some models such as MARBERT include dialectal Arabic (DA) in their training data. The following Arabic transformers have been proposed in existing literature to fine-tune several tasks, such as sentiment analysis, text generation, and QA.

*AraBERT* The AraBERT model, based on BERT, was proposed by *Antoun, Baly & Hajj (2020c)* and trained to process Arabic texts. The model used "Farasa" for segmentation and "SentencePiece" replaced "WordPiece" tokenizer to achieve more accurate results while processing Arabic text and increase the number of unique word-representations in a limited vocabulary. To clean the data, diacritics and elongation were removed, leaving the English characters. The training size was set to 700 GB of storage vocabulary (*Antoun, Baly & Hajj, 2020c*). The model was pre-trained to be used primarily for sentiment analysis for classification, named-entity recognition, and QA systems. AraBERT has the same architectural components as the BERT-Base model, with 12 encoder layers, 12 attention heads, and 768 hidden units.

Additionally, AraBERT employed the same two tasks of pre-train phase as BERT, namely MLM and NSP, as shown in Fig. 16. The main differences between BERT and AraBERT are the embedding mechanisms and the use of different datasets. The BERT was trained on 3.3B words (800 M+2500 M) from English Wikipedia and bank corpus (*Devlin et al., 2019*). The AraBERT was pre-trained using 24 GB of text, while the BERT was trained using 13 GB of English text.

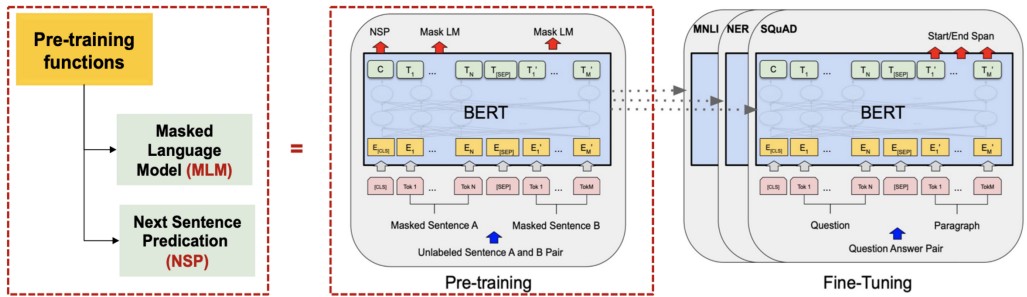

**Figure 16   Architecture of BERT and AraBERT (*Devlin et al., 2019*).**

*ARBERT and MARBERT*   Two deep bidirectional transformers for Arabic known as ARBERT and MARBERT was proposed by *Abdul-Mageed, Elmadany & Nagoudi (2021)*. Both transformer models were built and implemented based on the BERT transformer model. The models differed in the training data as these were trained on MSA data, social media data, and Arabic dialects. However, the models had the same stages as those of the BERT model. The ability of the models to handle different dialects of Arabic was improved by the augmentation of their training dataset by adding a set of randomly sampled 1B Arabic tweets.

*AraT5*   To build and train Arabic model based on text-to-text transfer transformer (T5) (*Raffel et al., 2020*) and mT5 (*Xue et al., 2021*) transformer models, Nagoudi et al. (*Nagoudi, Elmadany & Abdul-Mageed, 2022*) proposed and implement AraT5 for several tasks. The T5 was different from BERT because T5 had a casual decoder and used the task of fill-in-the-blank cloze, which replaced the masking task. AraT5 was trained on generating text such as translation machine, question generation, and summarization.

*ArabicBERT*   *Safaya, Abdullatif & Yuret (2020)* proposed pre-training of a BERT model using a concatenation of the Arabic version of OSCAR, a filtered subset from common crawl, and an Arabic Wikipedia dump totaling 8.2B words. This version of BERT was known as ArabicBERT.

*Arabic ALBERT*   A version of a lite BERT (ALBERT) (*Lan et al., 2020b*) designed for Arabic was proposed by Safaya which is available at https://ai.ku.edu.tr/arabic-albert/ (*Safaya, 2020*). The model was trained on data from the Arabic OSCAR corpus and Arabic Wikipedia.

*AraELECTRA*   Arabic-specific efficiently learning an encoder that classifies token replacements accurately (ELECTRA) was introduced by *Antoun, Baly & Hajj (2020b)*. The ELECTRA framework has two main components: the generator and discriminator. The discriminator is typically tuned for downstream tasks. Similar to AraBERT, AraELECTRA was trained on 77 GB of text data. The primary focus of the model was to identify "real" input tokens from "fake" ones generated by a separate neural network (*Clark et al., 2020b*). AraELECTRA uses a hidden layer size of 256, which is different from other models that have 768 neurons per layer.

*mBERT* Multi-lingual BERT (mBERT) was introduced in 2019 by Google research for several languages, such as the Arabic language (*Pires, Schlinger & Garrette, 2019b*). The mBERT model was pre-trained on a large corpus of Wikipedia texts from 104 languages. The corpus was processed using "WordPiece" tokenizer and introduced a vocabulary of 110K words, which allowed cross-lingual comparisons and applications.

*XLM-RoBERTa (XLM-R)* Multi-lingual extension of the original robustly optimized BERT pre-training approach (RoBERTa) model (*Liu et al., 2019*) was proposed by Conneau et al. (*Ruder, Søgaard & Vulic, 2019*). The model was trained on 2.5 TB of common-crawl data on 100 different languages.

*GigaBERT* *Lan et al. (2020a)* introduced the GigaBERT model, which was trained on a large set of Arabic news articles in order to improve its accuracy in predicting the sentiments of future articles. Training data was enhanced with English translations to improve performance across languages.

*CAMeLBERT* The CAMeLBERT corpus comprised a mixture of MSA, dialectal, and classical Arabic texts, totaling 17.3 B tokens. The model was proposed by *Inoue et al. (2021)* in 2021. Tokens used data to pre-train the CAMeLBERT-Mix model, while the model was fine-tuned by a limited number of Arabic NLP tasks and GigaBERT.

*QARiB* QCRI Arabic and dialectal BERT (QARiB) was proposed by *Abdelali et al. (2021)*. The model was trained on a combination of Arabic Gigaword, various sources of data including news articles, and 440 M tweets collected between 2012 and 2020.

*ArabicTransformer* The ArabicTransformer model was proposed by *Alrowili & Shanker (2021)*. The model is composed of an ELECTRA -objective and a funnel transformer. The funnel transformer was implemented to minimize the number of hidden states within a sequence by employing a pooling technique, resulting in a significant decrease in pre-training costs. The complete sequence of hidden states of the encoder was compressed into multiple blocks using the pooling technique. The ArabicTransformer model employs a B6-6-6 architecture, which includes three blocks, each with six layers and a hidden size of 768. Additionally, another model with a B4-4-4 design comprises three blocks, each with four layers, and a hidden size of 768. These models were trained on various tasks, including QA.

*hULMonA* *ElJundi et al. (2019)* introduced the first universal language model in Arabic (hULMonA) for pretrained language modeling, based on mBERT. The hULMonA was especially proposed for Arabic sentiment analysis tasks. The model was trained on 600 K Arabic articles from Wikipedia, with three hidden layers.

*AraGPT2* AraGPT2 is a stacked transformer-decoder model trained using the CLM objective, which was developed by *Antoun, Baly & Hajj (2020c)*. The model was trained on 77 GB of Arabic text and is the largest Arabic language model to date. The model is designed

to generate human-like text, allowing engagement in natural-sounding conversations. It can be used for a variety of tasks, such as text summarization and natural language generation.

A comparative list of Arabic PLMs is presented in Table 7, based on the year of development, reference, pre-training tasks, model architecture, corpus and vocabulary list size, type of text to build the corpus, and number of tokens and parameters. The comparison also includes the type of segmentation. Segmentation refers to the process of breaking-down words into their individual prefixes, stems, and suffixes (*Abdelali et al., 2016*). For example, "وكتابنا" meaning "and our book" can be segmented using "Farasa" to generate "و + كتاب + نا." Segmentation was proven to have a significant impact on NLP applications, such as IR (*Abdelali et al., 2016*). An example of most common Arabic segmenter is "Farasa," which is an Arabic NLP toolkit serving numerous tasks, one of which is segmentation of Arabic words.

A summary of evaluation results using Arabic PLMs for Arabic QA tasks is presented in Table 8. All models were Arabic PLMs except three models proposed for multi-lingual PLMs, which included mBERT (*Pires, Schlinger & Garrette, 2019a*), XLM-R (*Conneau & Lample, 2019*), and GigaBERT (*Lan et al., 2020a*). Upon observation of data presented in Table 8, Figs. 17 and 18, TyDiQA dataset results provided better evaluations when compared to other Arabic QA datasets. The TyDiQA dataset was of high quality owing to its cleanliness and accurate labeling, which was attributable to its development by Arabic language experts (*Alsubhi, Jamal & Alhothali, 2022*). However, the AQAD dataset consistently yielded poor results when compared to other datasets because the quality of the dataset played a significant role in determining the performance of models and the size of datasets (*Alrowili & Shanker, 2021*). The sub-optimal performance of ARCD and Arabic-SQuAD was attributed to the inferior quality of their training samples, which were translated from the English SQuAD (*Alsubhi, Jamal & Alhothali, 2022*). Additionally, the ARCD training dataset included text in languages other than Arabic, which could have led to a decrease in performance owing to the presence of unfamiliar sub-words and characters. Moreover, the size of training datasets was not large enough as compared to English QA datasets. This may have affected their performance (*Alsubhi, Jamal & Alhothali, 2022*). Conversely, the ArabicTransformer model (*Alrowili & Shanker, 2021*) yielded better results due to its large parameter size than those of other architectures such as AraELECTRA. However, the number of parameters and tokens were not presented.

### Taxonomy of Arabic QA technique

Before addressing the limitations of the included primary Arabic QA studies, the classification of the systems, models, techniques, datasets, domains, and approaches based on the Arabic QA tasks is presented in Fig. 19. All techniques, domains, approaches, and datasets illustrated and discussed in previous sections were included. The RB approach, shown in Fig. 19, was classified based on the three main components of common systems, excluding the DL approach. Each component contained examples of used Arabic NLP tools and techniques. Moreover, examples were presented with datasets categorized based on domain and type of questions.

Alrayzah et al. (2023), *PeerJ Comput. Sci.*, DOI 10.7717/peerj-cs.1633

**Table 7    Comparison of Arabic PLMs.**

| Ref. | Name | Pretraining tasks | Architecture | Corpus size | Type of text | Segmentation | Vocab size | #Tokens | #Params |
|---|---|---|---|---|---|---|---|---|---|
| *Pires, Schlinger & Garrette (2019b)*, *Mozannar et al. (2019)* | mBERT | MLM | Encoder | 1.4 GB | Arabic Wikipedia 2018 (MSA) | X | 110K | 153M | 110M |
| *ElJundi et al. (2019)* | hULMonA | MLM NSP | Encoder | N/A | Arabic Wikipedia articles up to March of 2019 | MADAMIRA (Arabic morphological analyzer and splitter) | N/A | 108M | N/A |
| *Conneau & Lample, (2019)* | XLM-R | CLM MLM | Encoder | 5.4 GB | Arabic Wikipedia 2018 (MSA) | X | 250K | 2.9M | 270M |
| *Antoun, Baly & Hajj (2020a)* | AraBERT | MLM NSP | Encoder | 23 GB | MSA | Farasa | 64k | 2.5B | 136M |
| *Antoun, Baly & Hajj (2020b)* | AraELECTRA | RTD NSP | Encoder | 77 GB | MSA | Farasa | 64K | 2.5B | Discriminator: 136M, Generator: 60M |
| *Safaya, Abdullatif & Yuret (2020)* | ArabicBERT | MLM NSP | Encoder | 95 GB | MSA & DA | X | 32k | 2.8B | 110M |
| *Abdul-Mageed, Elmadany & Nagoudi (2021)* | ARBERT/ MARBERT | MLM NSP/ MLM | Encoder | 61/128 GB | MSA/ MSA & DA | X | 100K/ 100K | 6.2B/ 15.6B | 163M/ 163M |
| *Antoun, Baly & Hajj (2020c)* | AraGPT2 | CLM | Decoder | 77 GB | MSA | X | 64K | 8.8B | 135M |
| *Safaya (2020)* | Arabic-ALBERT | MLM SOP | Encoder | 33 GB | Arabic Wikipedia & OSCAR (MSA/DA) | X | 32K | 4.4B | 110M |
| *Lan et al. (2020a)*, *Lan et al. (2020b)* | GigaBERT | MLM NSP | Encoder | 42.4 GB | MSA | X | 50K | 4.3B | 125M |
| *Abdelali et al. (2021)* | QARiB | MLM NSP | Encoder | 127 GB | MSA | Farasa | 64K | 14B | N/A |
| *Inoue et al. (2021)* | CAMeLBERT | MLM NSP | Encoder | 167 GB | MSA, DA & CA | Heuristic-based sentence segmenter | 30K | 17.3B | 108M |
| *Nagoudi, Elmadany & Abdul-Mageed (2022)* | AraT5 | Seq2seq MLM | Encoder-Decoder | MSA: 70 GB DA: 127 GB | MSA & DA | X | 100K | MSA: 7.1B DA: 21.9B | 220M |
| *Alrowili & Shanker (2021)* | ArabicTransformer | RTD NSP | Encoder | 45 GB | MSA | X | 50K | N/A | N/A |

**Notes.**

Ref., reference; N/A, not available; MSA, modern stranded Arabic; DA, dialect Arabic; CA, classical Arabic.

**Table 8  Evaluation results of Arabic QA using Arabic PLMs.**

| Model | Dataset name | | | | | | | |
|---|---|---|---|---|---|---|---|---|
| | Arabic-SQuAD | | ARCD | | TyDiQA | | Qur'an QA 2022 (QRCD) | |
| | F1 | EM | F1 | EM | F1 | EM | F1 | EM |
| AraBERT | 60.60 | 36.35 | 61.2 | 30.1 | 82.70 | 65.47 | 55 | 34 |
| AraELECTRA | 56.85 | 39.33 | 71.22 | 37.03 | 85.01 | 73.07 | 49.99 | 23.10 |
| mBERT | 48.6 | 34.1 | 50.10 | 23.9 | 64.02 | 46.36 | 31 | 0.9 |
| ArabicBERT | 62.24 | 30.48 | 62 | 30 | 81.24 | 67.42 | 47 | 33 |
| ARBERT | 67.90 | 49.92 | 60.73 | 27.21 | 66.94 | 46.80 | 58.7 | 37.3 |
| MARBERT | 58.46 | 41.09 | 55.14 | 23.22 | 57.51 | 38.98 | 51.5 | 32.1 |
| Arabic-ALBERT | 61.33 | 30.91 | 61 | 31 | 80.98 | 67.10 | – | – |
| XLM-ROBERTa | 64.91 | 45.88 | 59.61 | 27.31 | 60.99 | 39.41 | – | – |
| ArabicTransformer B6-6-6 | – | – | 72.70 | 36.89 | 87.21 | 75.35 | 53.47 | 16.38 |
| ArabicTransformer B4-4-4 | – | – | 67.70 | 31.48 | 85.89 | 74.70 | – | – |

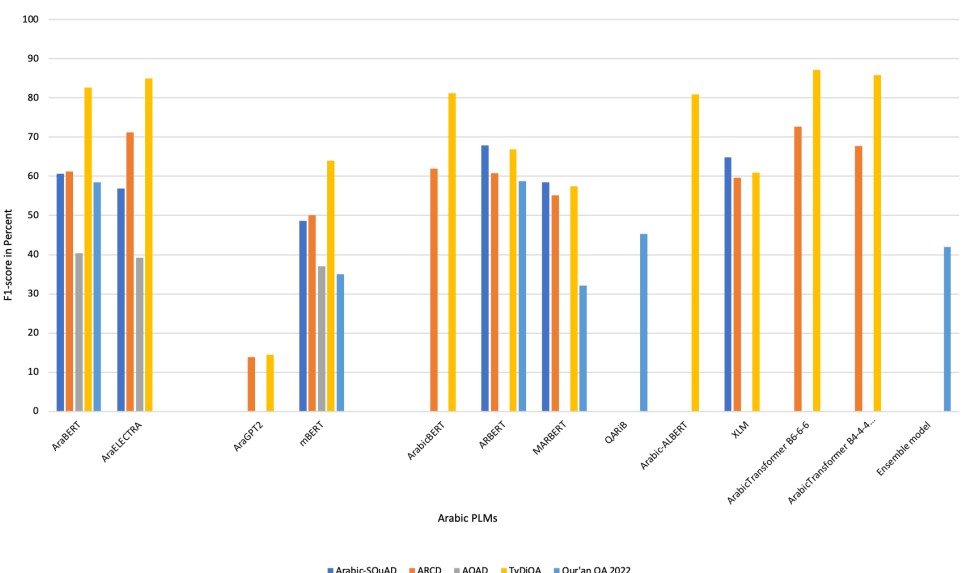

**Figure 17   Distribution of F1-score of Arabic QA based on PLMs.**

## RQ5: What are the limitations of Arabic QA studies?

Upon analysis of the included studies, numerous issues with several suggested models and systems were identified. Initially, a majority of the considered studies were based on RB approach until 2020. Thereafter, only seven studies developed Arabic QA models based on DL, while eight studies were already existing and used fine-tuning on the Holy Qur'an dataset. A majority of the Arabic PLMs models was only built for various NLP tasks such as sentiment analysis, test summarization, text classification, and

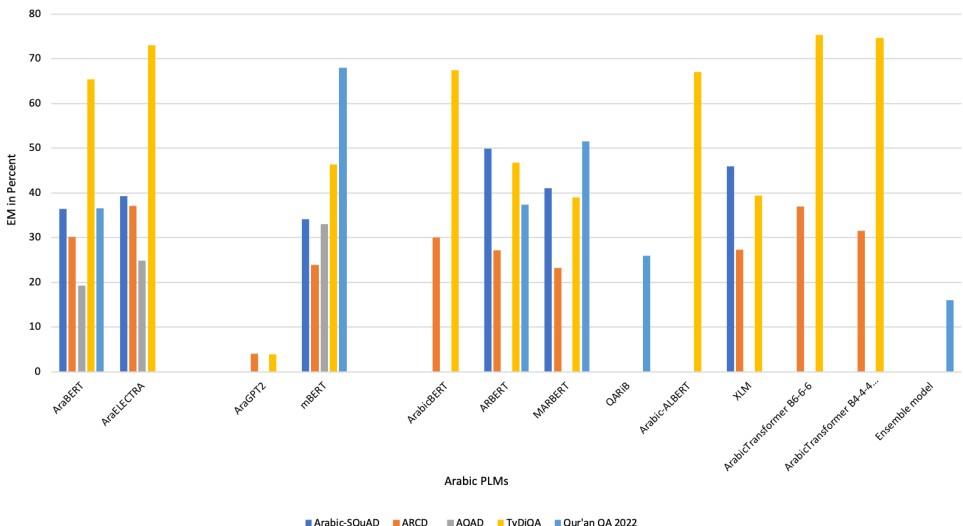

**Figure 18 Distribution of EM of Arabic QA based on PLMs.**

QA. Conversely, several English QA models were built for enhancing and improving reading comprehension of models for English QA, such as SpanBERT (*Glass et al., 2020*), ReasonBERT (*Deng et al., 2021*), SPLINTER (*Ram et al., 2021*), and BLANC (*Seonwoo et al., 2020*). A majority of such models achieved state-of-the-art results.

Contrarily, a majority of the proposed Arabic QA systems was suggested to enhance open-domain QA with 58% of all primary studies, as illustrated in Fig. 7. Researchers working on open-domain Arabic and QA systems focused on algorithms and techniques, used IR to retrieve documents. However, a little attention must be paid to answer extraction, which requires ability enhancement of machines for reading comprehension. Reading comprehension is an essential step for improving both closed and open domains to allow a model to return more accurate results.

Moreover, a majority of the datasets did not have large enough size as compared to other language QA datasets, such as SimpleQuestions (*Bordes et al., 2015*) and Wiki-Movies (*Miller et al., 2016*), produced for the English QA, which contain 108,442 and 100,000 QA pairs, respectively. The CLEF dataset was relatively small with 240 QA pairs, while the smallest dataset, TREC, only comprised 75 questions. These two datasets were mainly used for open-domain QA systems. The size of Arabic QA datasets affected model performance. Hence, larger sets were required. To fine-tune models on specific task and introduce accurate results, large dataset up to 100,000 samples were required because no exploitation of prior knowledge existed.

Conversely, approximately half of the used Arabic QA datasets were translated from English, which could have affected the quality of the used datasets, leading to poor performance. Moreover, primary studies (*Abdelnasser et al., 2014*; *Kamal, Azim & Mahmoud, 2014*; *Sadek, 2014*; *AlShawakfa, 2016*; *Sadek & Meziane, 2016*; *Shaker et al., 2016*; *Azmi & Alshenaifi, 2017*; *Shaker et al., 2016*; *Alamir et al., 2021*; *Maraoui, Haddar & Romary, 2021*) did not use a benchmark dataset because they created their own datasets,

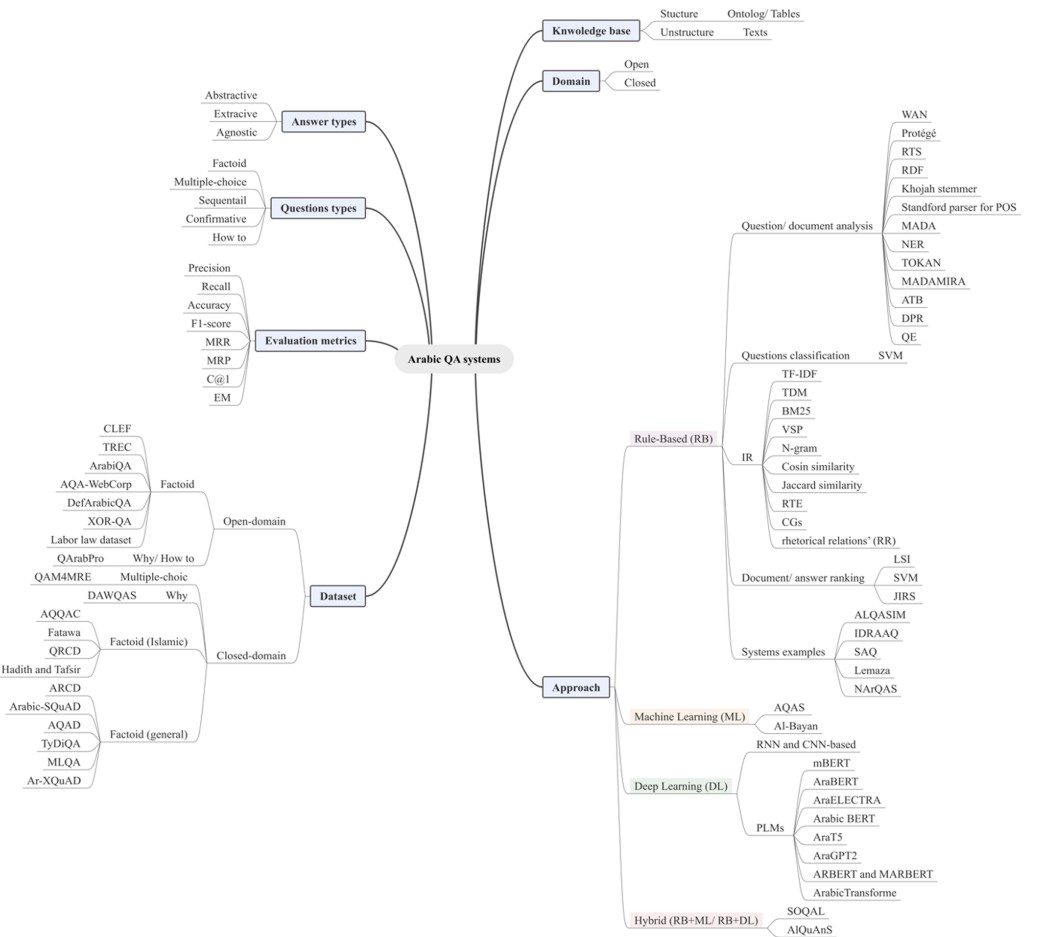

**Figure 19** Taxonomy of Arabic QA technique.

which were not published publicly. A majority of the created datasets was for the Islamic domain.

All studies based on ML (*Abdelnasser et al., 2014*; *Al-Shenak, Nahar & Halawani, 2019*) and hybrid (*Ahmed & Anto, 2016*; *Nabil et al., 2017*; *Ismail & Homsi, 2018*; *Mozannar et al., 2019*) approaches, classified questions using SVM only among other ML algorithms are listed in Table 2. None of the systems based on RB, ML, and hybrid approaches were available online. Therefore, no comparison could be drawn as they were evaluated on diverse datasets of differing sizes, with some lacking a benchmark dataset.

## RQ6: How to enhance the Arabic QA systems and models?

The limitations and issues in existing Arabic QA studies assisted in the identification of further research opportunities in the relevant area.

• Creation of large Arabic QA datasets that cover both specific domains and general open domains, making them available to all, is highly important. Moreover, the currently available Arabic QA datasets can be used by utilizing recent techniques of handling

scarce datasets. Examples of such techniques include data augmentation and few-shot learning (FSL). Data augmentation allows creation of new datasets by paraphrasing and using synonyms of the texts already existing in the datasets. Conversely, FSL is used to solve problems in machine learning. The problems include methods to train machines when data information are insufficient. Humans can recognize new object types from a tiny sample size, but most machine learning approaches require thousands of samples to recognize the objects (_Wang et al., 2020_). Recent successes in DL models rely on a large amount of data in several AI tasks such as image classification, machine translation, and speech modeling. However, obtaining enough samples for DL approaches can be challenging because data are often expensive or are infrequent (_Yan, Zheng & Cao, 2018_).

- Expanding closed-domain datasets to cover educational documents, books, and scientific papers.
- Training PLMs on the Holy Qur'an and Islamic corpus to enhance creating Islamic vocabulary, and then increasing the accuracy of the models.
- "Farasa," one of the tools available for handling Arabic text, can perform tasks such as text segmentation, lemmatization, and POS tagging.
- The answer extraction component can be enhanced by employing alternative techniques that not only display relevant sentences and paragraphs, but also generate or extract answers.
- Utilization of ML techniques for both question classification and answer extraction, or combining both ML and RB techniques, is advisable.
- Requirement for developing PLMs, especially for improving Arabic QA, as there are several English QA models achieving higher results in benchmark QA datasets, such as SpanBERT (_Glass et al., 2020_), ReasonBERT (_Deng et al., 2021_), SPLINTER (_Ram et al., 2021_), and BLANC (_Seonwoo et al., 2020_). A majority of such models have achieved state-of-the-art results.
- Enhancement of machine reading comprehension should be focused to increase the accuracy of returning the most accurate answers.

## LIMITATION

The findings of this systematic review were restricted to Arabic QA studies published between 2013 and 2022. Therefore, the analysis and interpretation of this review could be impacted if studies conducted prior to or after the considered era are included. The outcomes of this systematic review are dependent on the search terms, strings, and criteria of inclusion and exclusion that were selected during the search process. For example, existing studies based on non-factoid questions, such as Arabic community and chatbot systems, were excluded. Furthermore, only publications in the English language were considered, while studies published in other languages were excluded.

## CONCLUSIONS

This study offers a comprehensive analysis of existing literature on Arabic QA published between 2013 and 2022. Forty primary studies are selected using the Google Scholar

search engine. A variety of Arabic QA activities is also discussed. To investigate various factors that impact the effectiveness and applicability of the techniques proposed in the selected studies, a thorough analysis is conducted. The factors encompass datasets, domains, procedures, approaches, models, and evaluation measures. Furthermore, several difficulties and limitations of the listed studies are explored. Based on these constraints, new research directions are proposed. The discipline of Arabic QA is still in its infancy, especially when based on DL, and researchers have to face a variety of problems before gaining progress in the state-of-the-art. This systematic literature review is expected to provide extensive information on existing methodologies, techniques, datasets, and recent trends in Arabic QA, with the intention of inspiring further research in this field.

### Funding
The authors received no funding for this work.

### Competing Interests
The authors declare there are no competing interests.

### Author Contributions

- Asmaa Alrayzah conceived and designed the experiments, performed the experiments, analyzed the data, performed the computation work, prepared figures and/or tables, authored or reviewed drafts of the article, and approved the final draft.
- Fawaz Alsolami analyzed the data, authored or reviewed drafts of the article, and approved the final draft.
- Mostafa Saleh analyzed the data, authored or reviewed drafts of the article, and approved the final draft.

### Data Availability
   This is a literature review and did not utilize raw data.

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
