# Peer review of "Challenges and opportunities for Arabic question-answering systems: current techniques and future directions"

_PeerJ Computer Science, doi:10.7717/peerj-cs.1633_

## Round 0.1 · original submission · Major Revisions

Your manuscript requires major revisions before consideration for publication. Primarily, it is essential to clarify the system's objectives, challenges, novelty, and contributions in the introduction and elaborate on preprocessing tasks. A comprehensive chronological literature review, detailed representation of datasets, and system evaluation through precision, recall, and f-measure are needed. Address typographical errors, and restructure your research questions and the presentation of the selected articles for enhanced comparison and analysis.

Furthermore, your study would benefit from additional examination of the impact and novelty of the reviewed studies and potential benefits of meaningful replication in the field. Consistency in number representation is also necessary. We look forward to a revised submission of your manuscript addressing these matters.

·

Basic reporting

The language used in the review article is clear, unambiguous, and maintains a professional tone. The sentences are well-structured, and the vocabulary is appropriate for an academic review. The review demonstrates a high level of proficiency in the use of professional English.
The article thoroughly analyzes the literature, and 40 primary studies related to Arabic QA are considered in the systematic review. It also discusses various aspects of Arabic QA, including datasets, domains, procedures, approaches, models, and evaluation measures, which show a good level of field background/context.
The structure of the article is professional including the title, abstract, introduction, background, related work, survey methodology, results, limitations, conclusion, and references.
The review may be of interest to researchers in the field of NLP and Arabic language processing, it may not have a broad cross-disciplinary impact. Since the review article discusses various computational techniques like machine learning and deep learning, it falls in the scope of PeerJ Computer Science.
There is very limited work on Arabic QA systems. The review article provides useful insights in this domain. The article adequately introduces the subject, highlights the challenges and sets future directions. It also mentions that the review aims to investigate various factors that impact the effectiveness and applicability of the techniques proposed in the selected studies. The review article did not include any formal results, definitions of terms, theorems, or detailed proofs. Instead, being a review article it focused on the research questions, main findings, limitations, and suggested enhancements for Arabic QA systems.

Experimental design

Although the study design is well organized into a series of research questions, the first research question "Which are the Arabic QA studies conducted to date" should be covered in the related work. Moreover, instead of presenting summaries of all 40 primary selected articles, it would be better to compare these considering various aspects and key attributes/features of the studies.

Validity of the findings

The article successfully addresses the goals set out in the Introduction, presents a well-supported argument, and offers valuable insights into the challenges and opportunities for Arabic QA systems/models. However, it could benefit from further evaluation of the impact and novelty of the reviewed studies and the potential benefits of meaningful replication in the field.

Additional comments

Use one consistent style for numbers. they should be written either in English or in numeral form. e.g. see line 292
Correction needed. "as shown IN figure 13". line 989.

Reviewer 2 ·

Basic reporting

The Article is well written clear to understand and technically acceptable. Lots of similar work that was already done in many languages all over the world. Hence, automatic questions will generate then what is the objective of the system, what are the chal-lenges they face in Arabic, what is the Novelty of the system and what is (are) the contribu-tion(s). i.e. researchers wanted to contribute to this system?
Introduction must contain objective(s), Challenges, Novelty and Contribution of the system. 3. Researchers must elaborate preprocessing tasks because performance of the system depends completely with these preprocessing work, represent in tabular form with merits and demerits.

Experimental design

Literature Survey: (try scientific time travel for each language family) When the research on the said topic started. What others have proposed (discuss recent papers mostly, but keep some old and classic papers also). If possible, try to categorize the existing work. Say advantages and limitations of the past work (if per method it is not possible, then say something per category). Say in which category your work belongs to, and say why you have chosen this. Last paragraph should be pointing out research gaps (which would help you to set your objectives), and what you are proposing to address these gaps. For each category try to represent in tabular form.

Validity of the findings

Researchers must give examples of Data sets, represent in tabular form with merits and demerits.

Researchers needed to do the evaluation of the system with precision, recall and f-measure, represent in tabular form with merits and demerits.

They must perform a comparison with the similar system, represent in tabular form with merits and demerits.

Give analysis of the failures in the system with proper examples, represent in tabular form with merits and demerits.

Additional comments

9. Typo :
Line 50-54 and 171-175 are same (copy).

Annotated reviews are not available for download in order to protect the identity of reviewers who chose to remain anonymous.

---

## Round 0.2 · accepted · Accept

After reviewing the changes made, I confirm that the authors have taken all the reviewers' comments into account.

·

Basic reporting

No comments

Experimental design

No comments

Validity of the findings

No comments

Reviewer 2 ·

Basic reporting

1. Lots of similar work that was already done in many languages all over the world. Hence, automatic questions will generate then what is the objective of the system, what are the challenges they face in Arabic, what is the Novelty of the system and what is (are) the contribution(s). i.e. researchers wanted to contribute to this system?

2. Introduction must contain objective(s), Challenges, Novelty and Contribution of the system.

Answered: All are satisfied except the Novelty of the work is not clear.


9. Typo :
Line 50-54 and 171-175 are same (copy).

Answered: All are satisfied

Experimental design

3. Researchers must elaborate preprocessing tasks because performance of the system depends completely with these preprocessing work, represent in tabular form with merits and demerits.

Answered: All are satisfied.

4. Literature Survey: (try scientific time travel for each language family) When the research on the said topic started. What others have proposed (discuss recent papers mostly, but keep some old and classic papers also). If possible, try to categorize the existing work. Say advantages and limitations of the past work (if per method it is not possible, then say something per category). Say in which category your work belongs to, and say why you have chosen this. Last paragraph should be pointing out research gaps (which would help you to set your objectives), and what you are proposing to address these gaps. For each category try to represent in tabular form.

Answered: All are satisfied

5. Researchers must give examples of Data sets, represent in tabular form with merits and demerits.

Answered: All are satisfied



Answered: All are satisfied

Validity of the findings

6. Researchers needed to do the evaluation of the system with precision, recall and f-measure, represent in tabular form with merits and demerits.
Answered: All are satisfied
7. They must perform a comparison with the similar system, represent in tabular form with merits and demerits.
Answered: All are satisfied
8. Give analysis of the failures in the system with proper examples, represent in tabular form with merits and demerits.

Additional comments

Novelty of the work is not specified by the researchers. Say why are you doing similar work in your languages. what are the new unique things you may highlight in your language.